# Towards Understanding the Nature of Attention with Low-Rank Sparse Decomposition

**Zhengfu He**[1,2*]  **Junxuan Wang**[1,2*]  **Rui Lin**[1,2]  **Xuyang Ge**[1,2]
**Wentao Shu**[1,2]  **Qiong Tang**[2]  **Junping Zhang**[2]  **Xipeng Qiu**[1,2†]

[1]Shanghai Innovation Institute
[2]OpenMOSS Team, School of Computer Science, Fudan University

zfhe19@fudan.edu.cn

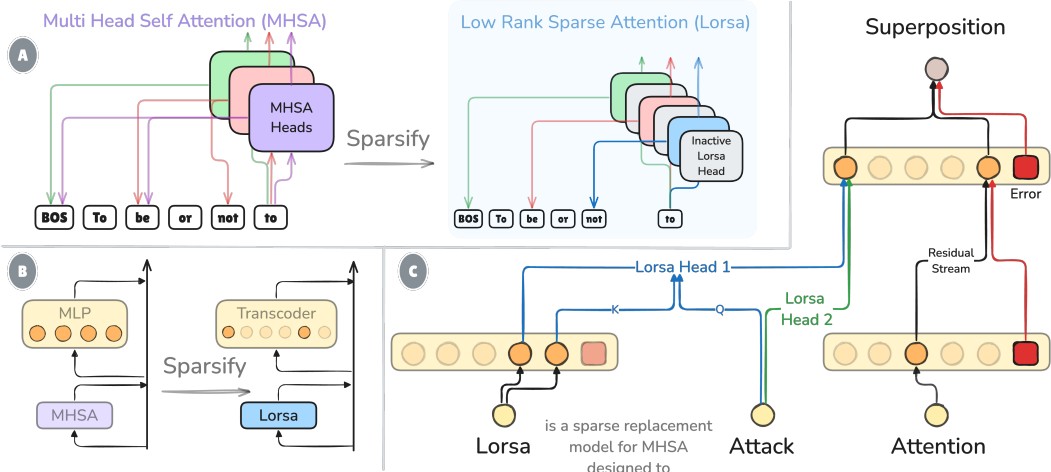

Figure 1: (**A**) Low-Rank Sparse Attention (Lorsa) comprises thousands of sparsely activated attention heads with 1D outputs, designed to extract interpretable attention units from the original Multi Head Self Attention (MHSA). (**B**) Lorsa serves as a replacement model for Transformer attention, substituting sparse interpretable components for attention modules. (**C**) Each Lorsa head explains an atomic feature-feature interaction across token positions, which was originally a part of an MHSA head or spread across multiple heads, i.e. put in attention superposition.

## ABSTRACT

We propose Low-Rank Sparse Attention (Lorsa), a sparse replacement model of Transformer attention layers to disentangle original Multi Head Self Attention (MHSA) into individually comprehensible components. Lorsa is designed to address the challenge of *attention superposition* to understand attention-mediated interaction between features in different token positions. Lorsa helps find cleaner and finer-grained versions of previously discovered MHSA behaviors like induction heads, successor heads, attention sink, and a comprehensive family of arithmetic-specific Lorsa heads. Interestingly, we identify a novel head type called *subtoken induction heads* that function at character level rather than token level. Automated interpretability analysis indicates that Lorsa achieves parity with SAE in interpretability while Lorsa exhibits superior circuit discovery properties. We also conduct extensive experiments on architectural design ablation, correlation to original MHSA heads and error analysis. Our early attempt to fully sparsify a toy Transformer succeeds to reveal clean global circuits. Eventually, we hope Lorsa

---

*Equal Contribution.
†Corresponding Author.

would help us greatly understand attention computation and enable full sparsifica­tion of model computation along with its MLP counterparts. Lorsa is open-sourced at `https://github.com/OpenMOSS/Language-Model-SAEs`.

# 1 INTRODUCTION

When examining the function of individual attention heads in a Transformer model, one might identify some of these heads implementing a specific behavior. A canonical example is induction heads which predicts 'Potter' following the token 'Harry' when 'Harry Potter' is present in the context (Olsson et al., 2022). Ablating these heads substantially prevents the model from correctly performing corresponding tasks, which indicates causal relation of these heads and the model's macroscopic behaviors. These interpretable attention units constitute the basic building blocks of the model's inter-token information mixing algorithm.

Not all attention heads, however, exhibit clear functionality. Most heads distribute attention across diverse contexts. Although some heads exhibit identifiable patterns, there might be inter-head collaboration that explains the whole story. These challenges in attention head interpretation is analogous to feature superposition in understanding individual neurons, which suggests the existence of **attention superposition** (Jermyn et al., 2024) in Multi Head Self Attention (MHSA), which we will further discuss in Section 2.

Inspired by the recent success of Sparse Autoencoders (SAEs) to extract monosemantic features from Transformers' hidden space (Templeton et al., 2024b) or approximate part of the network's computation as a sparse computation (Templeton et al., 2024a; Ge et al., 2024; Dunefsky et al., 2024), we propose Low-Rank Sparse Attention (Lorsa) to disentangle the atomic attention units from attention superposition (Section 3). Lorsa serves as a replacement module of the original MHSA with an overcomplete set of attention heads featuring a single-dimensional OV circuit (Elhage et al., 2021) and sparsity constraints.

We evaluate the reconstruction fidelity and sparsity trade-off of Lorsa in Section 4, along with scalability analysis. In Section 5, we introduce our exploration interface following Bricken et al. (2023), providing multifaceted information on each Lorsa head. We also quantitatively assess Lorsa head interpretability using top activations and their attribution patterns ($z$ pattern) with automated interpretability (Bills et al., 2023). The results indicate that Lorsa's monosemanticity is comparable to SAE features.

Section 6 presents findings with Lorsa on Pythia-160M (Biderman et al., 2023) and Llama-3.1-8B (Dubey et al., 2024). For validation, we first identify the Lorsa instantiations of known attention mechanisms: *induction heads*, *name mover heads* (Wang et al., 2023), *successor heads* (Gould et al., 2024), and attention sinks (Xiao et al., 2024). Furthermore, we characterize a family of arithmetic-specific Lorsa heads in Llama-3.1-8B. We also identify a subset of Lorsa heads in Llama-3.1-8B that function as *theme anchors* by exhibiting long-range, topic-specific attention patterns.

To the best of our knowledge, Lorsa is the first attempt to extract sparse interpretable attentional computation. We hope these findings facilitate future research (see Section 9 for limitations).

**Note on Terminology:** While prior work refers to the atomic computational units we aim to independently understand as *attentional features* (Jermyn et al., 2024; Ameisen et al., 2025), we adopt *attention units* to avoid conflating with activation-space features (which denote 1D linear features in representation spaces (Elhage et al., 2022)). The term *head* denotes MHSA or Lorsa heads as context dictates. We recommend viewing Lorsa as an interpretability tool rather than a drop-in replacement, since overparameterization may compound reconstruction errors.

# 2 ATTENTION SUPERPOSITION

Analogous to how post-ReLU neurons in Transformer MLPs learn to represent more features than they have dimensions (Elhage et al., 2022), a similar phenomenon may occur in Multi-Head Self Attention (MHSA). We hypothesize MHSA may comprise multiple attention units in **attention superposition**, each attending between certain token pairs with interpretable read/write operations on the residual stream. Under this hypothesis, we would expect (1) an atomic attention unit is spread

across multiple MHSA heads. (2) One MHSA head includes multiple units. We list three points of evidence of attention superposition in Transformer language models.

**1. A Few Neurons (Heads) Are Polysemantic.** Gurnee et al. (2023) discovered compound word neurons activating across diverse unrelated n-grams, while Bricken et al. (2023) reported neurons responding to mixed stimuli including academic citations and Korean text. (link). Similarly, successor heads (Gould et al., 2024) which increment 'Monday' into 'Tuesday' and '1' into '2' simultaneously exhibit Acronym behavior, Copying behavior and Greater-than behavior.

**2. Most Neurons (Heads) Exhibit Uninterpretable Activating (Attention) Patterns.** Multiple studies report the predominance of MLP neurons lacking clear activation patterns (Arora et al., 2018; Bricken et al., 2023). Likewise, Krzyzanowski et al. (2024) reports failed interpretation attempts for more than 90% heads in GPT-2.

**3. Attention Superposition in the Wild.** He et al. (2024a) and Kissane et al. (2024) both found attention output SAE features collectively contributed by multiple attention heads. If we consider SAE features to represent monosemantic directions, such distribution provides evidence for attention superposition. Furthermore, Jermyn et al. (2024) directly demonstrate this through a toy model where 5 ground-truth attention units are put in superposition over 2 attention heads. We also show that about 25% of our learned attention units are spread across multiple MHSA heads (Appendix E.2).

**Why Does Attention Superposition Matter?** Practically, attribution-based circuit tracing (Ge et al., 2024; Ameisen et al., 2025) becomes challenging when features are computed collectively: individual QK patterns do not explain the full mechanism and may be misleading due to interference from other features' computations within the same heads. The structure of attention superposition may relect intriguing motifs of model biology. For example, what makes some privileged attention units like induction heads mostly implemented by a single MHSA head (Olsson et al., 2022) while others are put in superposition? This parallels privileged bases in MLP neurons (Elhage et al., 2023).

## 3 Low-Rank Sparse Attention

### 3.1 Lorsa Architecture

---

**Algorithm 1:** Low-Rank Sparse Attention (~~MHSA~~ Lorsa)

---

**Input:** $\mathbf{X} \in \mathbb{R}^{n \times d}$: Input sequence (n tokens, d dimensions)
$W_q^h, W_k^h \in \mathbb{R}^{d \times d_h}$: Query/Key weights for head $h$. We adopt a QK sharing strategy so QK weights are not independent. See details below.
~~$W_v^h \in \mathbb{R}^{d \times d_h}$~~ $w_v^h \in \mathbb{R}^{d \times 1}$: 1-Dim Value weights
~~$W_o^h \in \mathbb{R}^{d_h \times d}$~~ $w_o^h \in \mathbb{R}^{1 \times d}$: 1-Dim Output weights
~~$H_{\text{MHSA}}$~~ $H_{\text{Lorsa}} \in \mathbb{Z}^+$: Number of Lorsa heads
$K \in \mathbb{Z}^+$: Max number of activated Lorsa Heads
**Output:** $\hat{\mathbf{Y}} \in \mathbb{R}^{n \times d}$: Output sequence

1 **for** $h \leftarrow 1$ **to** $H_{Lorsa}$ **do**
2     $Q^h = XW_q^h \in \mathbb{R}^{n \times d_h}$ ;        // Query projection for head $h$
3     $K^h = XW_k^h \in \mathbb{R}^{n \times d_h}$ ;         // Key projection
4     $v^h = Xw_v^h \in \mathbb{R}^{n \times 1}$ ;       // ~~$d_h$-Dim~~ 1-Dim Value projection
5     $A^h = \text{softmax}\left(\frac{Q^h(K^h)^T}{\sqrt{d_h}}\right) \in \mathbb{R}^{n \times n}$ ; // Attention patterns (Causal Mask)
6     $z^h = A^h v^h \in \mathbb{R}^{n \times 1}$ ; // ~~$d_h$-Dim~~ 1-Dimensional Weighted sum of values
7     $\hat{\mathbf{Y}}^{\mathbf{h}} = z^h w_o^h \in \mathbb{R}^{n \times d}$ ;        // Output of a single Lorsa head
8 $\mathcal{S} \leftarrow \text{TopKIndices}(\{z^h \mid h = 1, \ldots, H_{\text{Lorsa}}\}, K)$ ;    // Select top K heads by $z$
9 $\hat{\mathbf{Y}} = \sum_{h \in \mathcal{S}} \hat{\mathbf{Y}}^{\mathbf{h}}$ ;        // Add up ~~all~~ selected heads
10 **return** $\hat{\mathbf{Y}}$

---

We detail Lorsa's architectural designs in this section, with Algorithm 1 highlighting how Lorsa architecture differs from a standard MHSA layer. Lorsa takes in the same inputs of MHSA and is

trained to predict MHSA outputs. The training objective is simply minimizing the mean square error (MSE): $\mathcal{L} = \mathbb{E}_{\mathbf{x} \in \mathcal{D}} ||\text{Lorsa}(\mathbf{x}) - \text{MHSA}(\mathbf{x})||_2$.

**Rank-1 Output-Value Circuits.** Each MHSA head reads from and writes to a residual stream subspace via its OV circuit (Elhage et al., 2021), whose rank is decided by its head dimension $d_h$. Under the linear representation hypothesis that unidimensional features are encoded in the residual stream, we design Lorsa heads with rank-1 OV circuits. This offers the advantage of restricting read/write operations to one or few residual stream features (directions). Although ideal implementations would use rank-1 QK and OV circuits, we restrict dimensionality reduction to OV circuits for practical reasons.

**Query and Key Weights with Parameter Sharing.** We observe significant performance drop as rank of QK circuits $D_{\text{QK}}^{\text{Lorsa}}$ decreases, which is severer when $D_{\text{QK}}^{\text{Lorsa}} < D_{\text{QK}}^{\text{MHSA}}$. This may suggest QK circuits for attention units are multidimensional. In result, we choose $D_{\text{QK}}^{\text{Lorsa}} = D_{\text{QK}}^{\text{MHSA}}$ and implement parameter sharing for QK weights across every $G$ heads. Unless otherwise specified, we set $G = D_{\text{QK}}^{\text{Lorsa}}$ so that each head maintains a parameter count of $4D_{\text{model}}$ in average - equivalent to setting $D_{\text{QK}}^{\text{Lorsa}}$ to 1 without parameter sharing, which is crucial for Lorsa scalability.

Our parameter binding strategy renders Lorsa QK circuit strikingly similar to MHSA - a QK-sharing group of Lorsa heads is almost identical to an original MHSA head except the sparsity constraints applied on each OV dimension. We describe Lorsa heads as individual heads with shared QK circuits rather than a sparse dimension in MHSA architecture because they often exhibit correlated yet distinct interpretable functionalities, as we will show in Section 6. And there are cases where a QK-sharing group of Lorsa heads show no clear semantic correlation (Appendix C).

We also show in Appendix B.3 that Lorsa QK circuits are not solely learning to copy the original QK circuits. This distinguishes Lorsa from only applying sparse dictionary learning or Independent Component Analysis on OV circuits (Ameisen et al., 2024).

**Orders of Magnitudes More Heads and Sparsity.** To capture numerous underlying attention units, Lorsa employs an overcomplete architecture with $H_{\text{Lorsa}} \gg H_{\text{MHSA}}$ heads per layer, activating only $K \ll H_{\text{Lorsa}}$ heads per token. This parallels learning more features than the input dimension while enforcing sparsity in SAEs.

For a given token position, Lorsa's output aggregates the Top-K heads with largest $z$'s, where $z$ is the scalar activation value of a Lorsa head[1]. The active head subset dynamically varies across token positions. This sparsity mechanism resembles TopK-SAEs (Gao et al., 2024), as both select the $K$ most salient linear components.

**Connection to Sparse Autoencoders.** Lorsa shows notable resemblance to attention SAEs (Kissane et al., 2024) for its rank-1 OV circuits. Lorsa learns an overcomplete linear basis of the attention output space $\{w_o^h \mid h = 1, \ldots, H_{\text{Lorsa}}\}$ with sparsely activated scalar components $\{z_i^h \mid h = 1, \ldots, H_{\text{Lorsa}}\}$ at the $i$-th position, which is analogous to SAE decoder and sparse feature activations.

However, whereas SAE features are computed via single linear encoders with ReLU, Lorsa head activation at a given position $z_i^h$ derives from attention patterns $A_i^h$ and $v^h$ of previous tokens. Moreover, SAEs take in and predict the same activations while Lorsa, like Transcoders (Ge et al., 2024; Dunefsky et al., 2024), learns to predict downstream activations. It is more similar to a Gated (Rajamanoharan et al., 2024) Transcoder taking in activations from multiple positions, where the QK circuit resembles the *gate* with a non-linearity and $w_v$ is simply a linear encoder.

## 3.2 LORSA TRAINING

The Low-Rank Sparse Attention modules we are studying throughout this work are trained on all layers of Pythia-160M and Llama-3.1-8B. The training data is sampled from 800 million tokens for

---

[1]Conceptually, a Lorsa head's activation on a sequence should be $z^h ||w_o^h||_2$ rather than $z^h$. For analytic simplicity and clarity, we construct a model with identical predictions but set $w_v^h \leftarrow w_v^h ||w_o^h||_2$, $b_v^h \leftarrow b_v^h ||w_o^h||_2$ and $w_o^h \leftarrow w_o^h / ||w_o^h||_2$. This operation isolates activation $z^h$ from output direction $w_o^h$.

each model. The prompts are collected from SlimPajama (Soboleva et al., 2023) truncated to 256 tokens for Pythia and 1024 tokens for Llama.

Best practices for Lorsa training (e.g. Adam optimizer, warm-stable-decay schedule, optimal lr scaling law, etc.) largely complies with ones adopted in Templeton et al. (2024b). Training one Lorsa module with settings described in Table 1 takes 2 Nvidia A100 GPU hours for Pythia (batch size = 4,096 tokens) and 24 hours for Llama (batch size = 16,384 tokens).

| Target Model | # Heads | | | | Head Dimension | | | # Active Heads per Token | | # Params Per Layer | |
| --- | --- | --- | --- | --- | --- | --- | --- | --- | --- | --- | --- |
| | MHSA | Independent Lorsa QK | Lorsa QK | Lorsa OV | MHSA | Lorsa QK | Lorsa OV | MHSA | Lorsa | MHSA | Lorsa |
| Pythia-160M | 12 | 96 | 6K | 6K | 64 | 64 | 1 | 12 | 64 | 2.25M | 18M |
| Llama-3.1-8B | 32 | 256 | 32K | 32K | 128 | 128 | 1 | 32 | 128 | 64M | 512M |

Table 1: Architectural setups for both target models. We primarily focus on Lorsa modules with 500-1,000 times more heads than the original MHSA. For instance, we have 6K Lorsa heads for an MHSA layer in Pythia-160M, with every $D_{\text{QK}}^{\text{Lorsa}} = D_{\text{QK}}^{\text{MHSA}} = 64$ heads sharing QK weights. This gives us 96 independent QK weights.

Both models adopt Rotary Embedding (RoPE) (Su et al., 2021) and Llama uses Grouped Query Attention (GQA) (Ainslie et al., 2023). We show how Lorsa fits these modifications in Appendix A.

## 4 EVALUATING LORSA FIDELITY-SPARSITY PERFORMANCE

### 4.1 $L(N, K)$ SCALING LAWS

We explore Lorsa scaling laws with respect to both number of learnable parameters $N$ and their sparsity $K$ (i.e. number of active Lorsa heads per token) as shown in Figure 2, compared to Top-K SAEs (Gao et al., 2024). Despite similar scaling trends, there is a notable gap between Lorsa and SAE under the same parameter budget and sparsity, especially when $K$ is large. Such comparison in terms of reconstruction fidelity and sparsity is in favor of SAEs since Lorsa learns QK and OV circuits to predict attention output with hundreds of activations, while SAE adopts a standard dictionary learning setting with the same input and output.

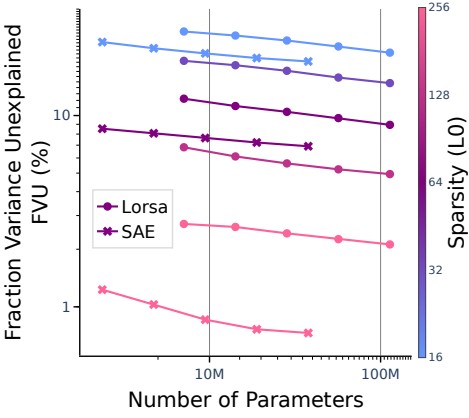

### 4.2 PER-LAYER EVALUATION

Figure 3 shows Lorsa's per-layer reconstruction error on Pythia-160M and Llama-3.1-8B in terms of fraction of variance unexplained (FVU).

Figure 2: Scaling laws of FVU against number of parameters and fixed L0 for SAEs and Lorsas trained on layer 3 in Pythia-160M.

We would like to highlight the notable correlation between trends of FVU across layers yielded by Lorsa and SAE in both models. We also observe strong correlation between these two sparse dictionary learning methods in terms of per-token error norm and direction (Appendix G).

## 5 ASSESSING LORSA INTERPRETABILITY

### 5.1 INTERPRETING INDIVIDUAL LORSA HEADS

**Top Activations.** With Lorsa heads' output restricted to a single direction, their activation strength at a given position $i$ can be described with a scalar $z_i^h$ (Section 3.1). Similar to SAE interpretation methods (Bricken et al., 2023; Templeton et al., 2024b), we iterate over 100M activations from a held-out dataset to identify the 16 highest-activating tokens for each Lorsa head.

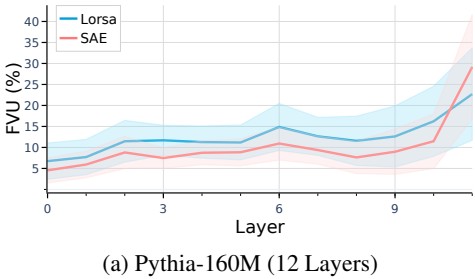

(a) Pythia-160M (12 Layers)

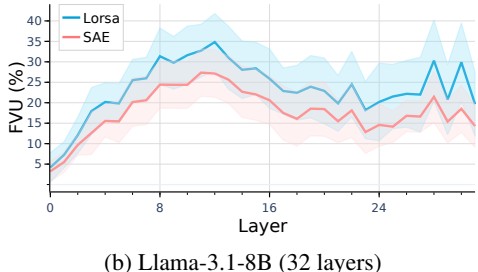

(b) Llama-3.1-8B (32 layers)

Figure 3: Per-layer reconstruction FVU for Top-K SAEs and Lorsas. All Pythia modules (left) comprises 18M learnable parameters and $K = 64$. Llama modules (right) have 512M parameters and $K = 128$. We evaluate the mean and standard deviation (shown as shaded areas) with 64K tokens.

$z$ **Pattern.** According to Algorithm 1, the top activations $z_i^h$ decompose linearly into token-wise contributions from preceding positions: $z_i^h = A_i^h v^h = \sum_{j=1}^{i} A_{i,j}^h v_j^h$, where $A_{i,j}^h$ denotes attention weight from token $i$ to token $j$ and $v_j^h = w_v^h \mathbf{x}_j$. Conceptually this tells from which previous tokens the activation $z_i^h$ is computed. Thus we call it the $z$ pattern. This is analogous to direct feature attribution (DFA) analysis for attention SAEs (Kissane et al., 2024; He et al., 2024a). An SAE feature's activation at the $i$-th token $f_i$ can be decomposed along heads and sequence position, i.e., $f_i = \sum_{j \leq i} \sum_{h \in H} W_f^{\text{enc}} o_j^h$, where $o_j^h$ is a linear component of MHSA output at token $j$ from head $h$. The DFA from token $j$ is then defined as $\sum_{h \in H} W_f^{\text{enc}} o_j^h$. In comparison, Lorsa's attribution includes only one rank-1 OV circuit and a single, though shared, QK circuit without multi-head aggregation. This enables QK circuit attribution for attention units distributed across multiple MHSA heads.

## 5.2 VISUALIZATION INTERFACE

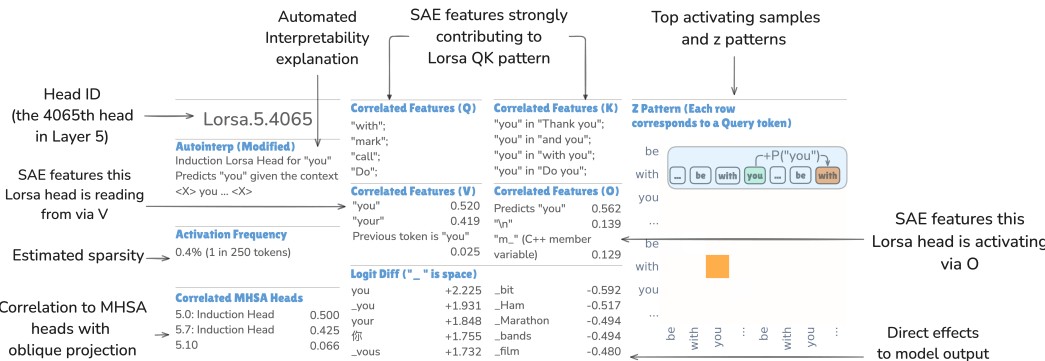

Figure 4: Visualization dashboard for a "you"-specific induction Lorsa head. We provide an example interpretation of each item below.

Our visualization interface provides multifaceted information on Lorsa head interpretation. We illustrate our dashboards with the example in Figure 4, which visualizes to an induction Lorsa head specifically firing for the token "you". The methods used to identify correlated MHSA heads and SAE features are described in Appendix E and F.

- **Correlation to SAE features / Logits via OV:** It mainly reads from *current token is "you"/"your"* features via its $w_v^h$; It strongly activates a *say "you"* feature (i.e., a feature amplifying the logit of "you" via the logit lens (nostalgebraist, 2020)); It amplifies the logits of a variety of "you" tokens.

- **Correlation to SAE features via QK:** Its QK attention pattern is mainly computed by *current token is "X"* features on the query position and *previous token is "X" & current token is "you"* features on the key side, where "X" can be a number of tokens that often precedes "you", such as "with", "thank" or "do".

- **Correlation to MHSA heads:** This Lorsa head is almost equally distributed in MHSA.5.0 and MHSA.5.7. Both MHSA heads exhibit induction functionality, as shown in Appendix E.

## 5.3 QUANTITATIVE EVALUATION WITH AUTOMATED INTERPRETABILITY

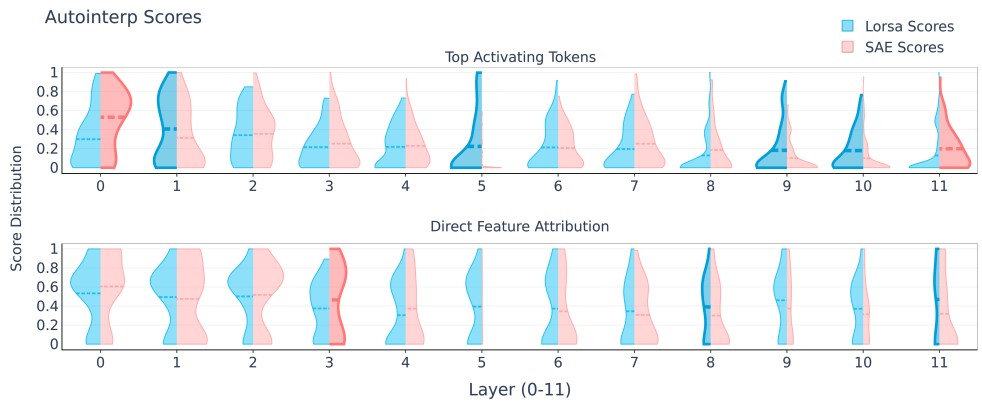

Figure 5: Automated interpretability scores of Lorsa heads and SAE features. Each distribution is estimated with 100 heads / features. The average score of each group is represented by a horizontal dash line. We highlight distributions with larger mean value suggested by t-tests with $\alpha = 0.05$.

To quantify the interpretability of Lorsa heads in terms of its top activations and $z$ pattern, we perform automated interpretability (autointerp) (Bills et al., 2023) with GPT-4o to estimate how comprehensible each Lorsa head is. We apply standard autointerp on max activating samples and extend to Lorsa $z$-patterns and direct feature attribution of attention output SAEs (Kissane et al., 2024). Prompt design, scoring method and choice of few-shot examples are detailed in Appendix I. All results are obtained with Pythia-160M Lorsa and SAEs of the same size.

As shown in Figure 5, Lorsa achieves a higher score in 6 cases, with 3 losses and 15 ties at $\alpha = 0.05$ significance across 24 layer-wise comparisons, suggesting comparable interpretability to SAE features. Both methods exhibit descending scores in deeper layers. Potential explanations include: (1) increased polysemanticity in later layers, or (2) limited capacity of current autointerp pipelines to capture long-range dependencies.

## 6 SEARCHING FOR SPECIFIC LORSA HEADS

We use path patching (Wang et al., 2023; Conmy et al., 2023) to find the Lorsa heads involved in specialized tasks. For a given Lorsa head, path patching ablates its output and allows the influence to propagate only through residual connections and MLPs (but not through other attention heads). This measures the head's counterfactual influence on the model's behavior.

### 6.1 LORSA RE-DISCOVERS PREVIOUSLY REPORTED HEADS

Previous works have documented attention heads with specific functionalities in well-characterized contexts (Section 7.1). We demonstrate that Lorsa rediscovers more specialized units of these attention behaviors due to its rank-1 OV circuit. Lorsa also isolates an important phenomenon called attention sink (Xiao et al., 2024) from other semantically meaningful heads. Figure 6 showcases four such heads, with their visualization dashboards provided in Appendix D.2. A representative selection of interpretable Lorsa heads is presented in Table 2.

We want to highlight an interesting variant of induction heads we call subtoken induction heads where the prediction operates at the subtoken level. When the sequence contains "[ Marion] ... [M]", the head predicts "[arion]", despite involving three distinct tokens ([A] [B] ... [C]). This occurs because the leading space in "[ Marion]" causes tokenization misalignment, splitting what would otherwise be a single token into subcomponents.

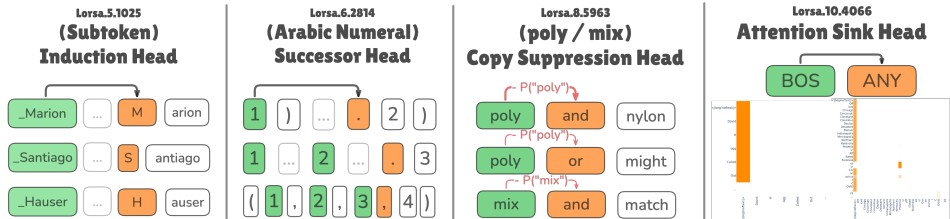

Figure 6: Examples of Lorsa heads re-discovering **finer-grained or cleaner versions** of previously reported heads. **Lorsa.5.1025**: A subtoken induction head for names, see details below. **Lorsa.6.2814**: A successor head attending to the previous arabic numeral token (almost exclusively 1, 2, and 3) and predicts its successor. **Lorsa.8.5963**: A copy suppression head attending to the previous token (almost exclusively 'poly' and 'mix') and suppresses its copy. **Lorsa.10.4066**: An attention sink head almost exclusively attending to the '<|beginoftext|>' token.

| Lorsa Head ID | Manual Interpretation |
|---|---|
| Lorsa.5.3955 | Induction for "ve" |
| Lorsa.5.4010 | Induction for last names |
| Lorsa.7.4203 | Induction for abbreviations |
| Lorsa.9.132 | Induction after "and"/"with" |
| Lorsa.9.1622 | Induction in Italian |
| Lorsa.4.32 | "define"/"include" in PHP |
| Lorsa.4.3013 | "public static" in Java |
| Lorsa.5.4035 | Say "Four"/"Five" |
| Lorsa.8.142 | Apple Inc. and products (iPhone etc.) |
| Lorsa.4.5167 | Previous token is "can"/"could" |
| Lorsa.11.6084 | Previous token is "make" |
| Lorsa.4.487 | Abbreviations (parentheses/quotes) |
| Lorsa.6.1491 | Abbreviations in parentheses |
| Lorsa.6.1787 | Abbreviations in parentheses |
| Lorsa.6.5499 | Abbreviations in parentheses |
| Lorsa.4.1420 | Russian contexts |
| Lorsa.9.1622 | Induction in Italian |
| Lorsa.4.4388 | Attention sinks |
| Lorsa.7.862 | Attention sinks |
| Lorsa.6.2592 | "the other"/"another" |
| Lorsa.10.1232 | Year of birth and death |

Table 2: A non-exhaustive collection of interpretable Lorsa heads we have found, which are grouped by color from top to bottom: induction heads, specific token heads, previous token heads, acronym heads, language-specific heads, attention sink heads, and miscellaneous heads.

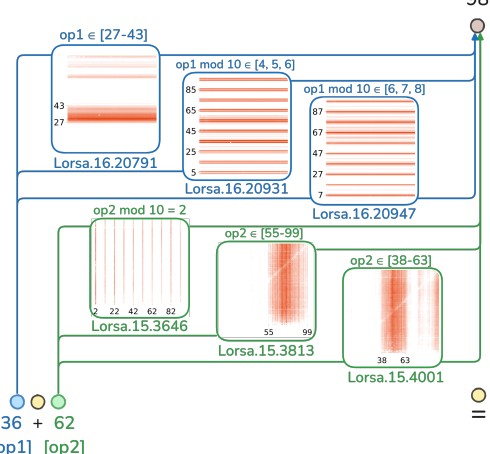

Figure 7: For the prompt "36 + 62 =", Lorsa moves two operands to the last position with 3 heads each. The first operand (36) is attended in terms of $z$ pattern by an "op1 $\in 27 - 43$", an "op1 % 10 $\in [4, 5, 6]$" and an "op1 % 10 $\in [6, 7, 8]$" head, which uniquely determines "op1 = 36". The same applies to op2.

## 6.2 A FAMILY OF ARITHMETIC LORSA HEADS IN LLAMA-3.1-8B

We identify a group of arithmetic-specific Lorsa heads in Llama-3.1-8B that activate during simple arithmetic operations following the template [op1][operator][op2][=]. One observation is that each head fetches certain operands with a number of unrelated heuristics, consistent to prior findings at neuron level on arithmetic mechanisms (Nikankin et al., 2024), despite Lorsa's architectural differences.

Figure 7 demonstrates an example of the prompt "36 + 62 =". Similar to Ameisen et al. (2025), we visualize the function of each Lorsa head with an operand plot, displaying its activity on the 100 × 100 grid of potential inputs of the template "op1+op2=".

These six Lorsa heads exhibit consistent interpretations in terms of their operand plots and $z$ patterns sampled from natural language prompts like "The price went up by 27% from $100 to". We exemplify this in Appendix D.3, along with more examples of arithmetic-specific Lorsa heads. We also conduct very preliminary pertubation experiments in arithmetic tasks to validate Lorsa's causal influence on the model's behavior, as described in Appendix D.4.

### 6.3 Lorsa Heads as Theme Anchors

While exploring through Lorsa heads in Llama-3.1-8B, we notice a distinctive subset of Lorsa heads attending to keywords with remarkable theme consistency from all subsequent tokens in a sentence. Figure 12 in Appendix D.5 illustrates two representative cases which exhibit relatively selective, long-range attention to tokens related to *presidency* and *dynamical systems* as evidenced by $z$ pattern. Through manual inspection we also find Lorsa heads activating on topics like alcohol addiction, dynamic system, medication instructions and terms of service.

These heads may serve as *theme anchors* maintaining topic representations to bias predictions. They may relate to SAE features "smeared" across token positions, as mentioned in Lindsey et al. (2025) (link) (example).

## 7 Related Work

### 7.1 Explaining Individual Attention Heads

With the help of activation patching (Meng et al., 2022; Zhang & Nanda, 2024) or path patching (Wang et al., 2023; Conmy et al., 2023), the literature has discovered a number of heads that exhibit certain functionality in pre-defined contexts. This line of research starts from a composition of *previous token heads* and *induction heads* (Olsson et al., 2022) which is closely related to in context learning. More work on this line includes *name mover heads* (Wang et al., 2023), *number comparison heads* (Hanna et al., 2023), *copy suppression heads* (McDougall et al., 2023), *successor heads* (Gould et al., 2024) and *long context retrieval heads* (Wu et al., 2024).

### 7.2 Superposition Hypothesis and Sparse Autoencoders

The superposition hypothesis (Arora et al., 2018; Olah et al., 2020; Elhage et al., 2022) assumes that neurons are related to multiple non-orthognal underlying features. Sparse Autoencoders (Cunningham et al., 2023; Bricken et al., 2023) are proposed to extract an overcomplete set of the sparse and linear comprehensible features. Importantly, the success of the technique also sheds light on universality of superposition across model size (Templeton et al., 2024b; Lieberum et al., 2024; He et al., 2024b), model architectures (Wang et al., 2024) and modality (Abdulaal et al., 2024).

### 7.3 Sparse Autoencoder Variants

SAEs have developed multiple variants. Some improve initialization (Conerly et al., 2024), loss function (Conerly, 2024; Bussmann et al., 2024) or sparsity constraints (Gao et al., 2024) to solve specific issues such as shrinkage (Wright & Sharkey, 2024) and massive inactive features (Bricken et al., 2023).

Another direction of improvement is the SAE architecture. For instance, Gated SAEs (Rajamanoharan et al., 2024) are proved effective in mitigating shrinkage. Transcoders (Ge et al., 2024; Dunefsky et al., 2024) aims to simplify sparse circuit analysis by replacing MLPs, whose non-linear nature makes causal attribution intractable.

## 8 Discussion and Limitations

We report intriguing findings and limitations. Despite promising results, key challenges remain for future work in the following aspects.

**Unbinding QK circuits.** A key limitation is shared QK circuits: heads may not be fully independent despite positive $z$ pattern findings, since $z$ mixes Q, K and V. Circuit tracing risks mis-attributing QK to components of other heads sharing the same circuit.

**Dynamically Reducing QK Rank.** One solution to unbind QK circuits is to reduce QK rank for each Lorsa head. If we could overcome the performance degradation of low-dimensional QK circuits, it is possible to scale up Lorsa with more independent QK circuits and fewer residual stream features

interacting via $QK^2$. This is also crucial for circuit tracing methods to have a clearer attribution of QK circuits with fewer features involved.

Moreover, our current design of Lorsa QK circuits assumes that all attention units have the same rank (i.e., $d_{\text{head}}^{\text{QK}}$). In Appendix C we show that Lorsa QK rank can be varied across heads by visualizing the singular values of $W_Q$ and $W_K$. A mechanism to dynamically determine the rank of QK circuits for each Lorsa head would be a promising direction for future work.

**Dark Matters.** We find non-trivial correlation between Lorsa error and SAE errors trained on the same attention layer in terms of (1) average loss per layer (2) loss per token on the same context and (3) error direction, as shown in Appendix G. This may suggest the existence of universal dark matters (Olah & Jermyn, 2024; Engels et al., 2024) for sparse dictionary learning methods like SAE and Lorsa. Any progress along this direction to reduce or understand SAE / Lorsa dark matters should reveal many interesting behaviors of neural networks.

**Inactive Attention SAE Features and Lorsa Heads.** Despite efforts on hyperparameter search, we find that attention SAE and Lorsa both contains a majority of inactive feature / heads (i.e. not activated once in 1e6 tokens). This phenomenon renders most computation wasted and raises a question about the difference between structure of attention output space and MLP output space or residual streams, where SAEs of the same size only have few dead features if configured properly.

**Cross Layer Attention Superposition.** If certain inter-token feature interaction is performed in more than one layer, our current method which decomposes only one MHSA layer does not suffice to find such relation. This parallels the problem of cross-layer superposition (Templeton et al., 2024b) for residual stream features. A cross-layer variant of Lorsa (Lindsey et al., 2024) might be tractable.

**Global Weights and Systematic Q/K/V Composition.** To better understand the global attention behavior of Transformers, one important research direction is to identify systematic Q/K/V composition like induction heads and previous token heads. Since Lorsa reveals finer-grained versions of MHSA heads, we can expect to find more of such cross-layer collaboration behavior. However, we failed in our early attempts to find Lorsa heads with Q/K composition.

## 9 CONCLUSION

In this work, we introduced Low-Rank Sparse Attention (Lorsa) to disentangle atomic attention units from attention superposition in Transformer models. Our experiments validated that Lorsa can recover known attention mechanisms and uncover novel interpretable behaviors. The scalability and quantitative autointerp results suggest the potential of Lorsa to adapt to real-world applications, especially unveiling the nature of attention computation in systematic end-to-end circuit tracing.

We hope Lorsa will eventually enable full sparsification of model computation alongside MLP counterparts. Our initial attempt on a two-layer Transformer unveils a clean induction circuit (Appendix H).

## ACKNOWLEDGMENTS

This work was supported by the National Natural Science Foundation of China (No. 62525602).

---

[2]It might also be the case that attention units must be described in multidimensional QK circuits, like induction heads requiring attending to multiple "the previous token is X" features.

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

# A  APPLYING LORSA TO MHSA VARIANTS

Modern transformer-based models commonly employ variants of multi-head self-attention (MHSA), such as those incorporating rotary position embeddings (RoPE) (Su et al., 2021) and grouped-query attention (GQA) (Ainslie et al., 2023). Lorsa demonstrates compatibility with these MHSA variants through straightforward adaptations.

- For RoPE-based MHSA layers, we apply the same rotary transformations to Lorsa's computed queries and keys before computing attention scores, maintaining the positional information encoding.
- In GQA implementations, Lorsa operates without modification—specifically, we intentionally avoid introducing grouped queries within the Lorsa framework.

Empirical results on both Pythia-160M and Llama-3.1-8B demonstrate that this design choice does not adversely affect performance. We apply these architectural variants based on the TransformerLens library (Nanda & Bloom, 2022).

# B  ABLATION STUDY ON CRUCIAL ARCHITECTURAL DESIGNS

We conduct ablation studies on two crucial architectural designs: (1) the query and key dimension and (2) the binding ratio. Our experiments validate the necessity of maintaining both the QK dimension and the binding mechanism in our proposed architecture. Additional ablation tests on other implementation details further validate our decisions.

Furthermore, we derive two **hard constraints** for parameter selection (violating these constraints leads to significant performance degradation):

- The QK dimension must not be smaller than the head dimension in MHSA
- The number of QK pairs must not be fewer than the number of attention heads in MHSA

## B.1  ABLATION STUDY ON QK DIMENSION

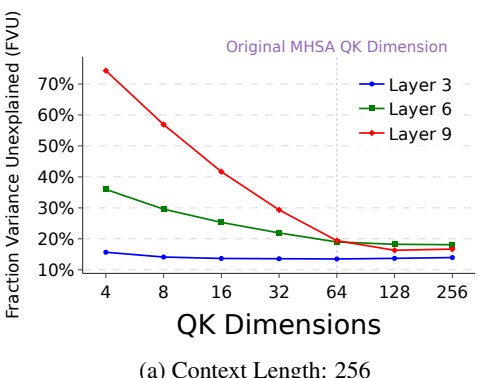 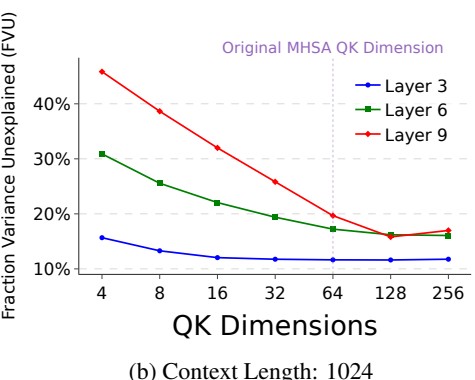

(a) Context Length: 256                         (b) Context Length: 1024

Figure 8: Ablation study on the QK dimension using Pythia-160M under different context lengths ($K = 64$). We fix the parameter budget across all settings and observe that reducing the QK dimension below the original MHSA head dimension ($d_{\text{head}} = 64$) results in significant performance degradation, highlighting the importance of maintaining a high QK dimension.

We conduct ablation studies on the QK dimension using Pythia-160M, evaluating performance under different context lengths (256 and 1024 tokens). To ensure fair comparison, we fix the parameter budget at $4D_{\text{model}}$ per attention head and maintaining a total parameter count equivalent to $4\times$ the original MHSA configuration throughout all experiments. As shown in Figure 8, reducing the QK dimension below the original MHSA's head dimension ($d_{\text{head}} = 64$) leads to severe performance degradation. This empirical evidence supports our design choice to maintain a high QK dimension.

## B.2 ABLATION STUDY ON BINDING RATIO

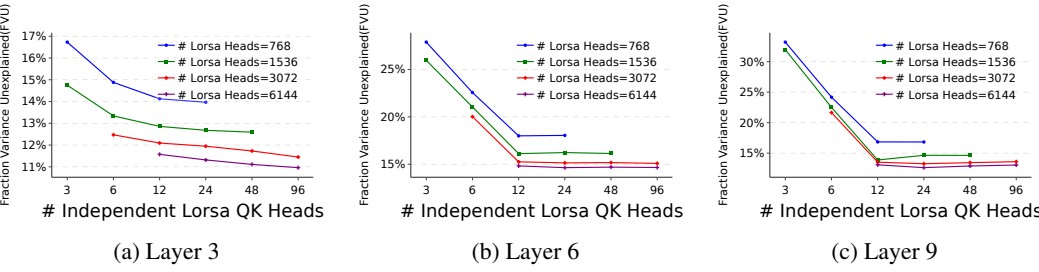

(a) Layer 3        (b) Layer 6        (c) Layer 9

Figure 9: Ablation study on the binding ratio. We vary the number of independent Lorsa QK heads and evaluate model performance under different settings. Appropriate binding maintains performance while reducing QK circuit cost, whereas overly aggressive binding (below the number of original MHSA heads) leads to substantial degradation.

We conduct a systematic study on the impact of the number of independent Lorsa QK heads (i.e., the number of Lorsa heads divided by the binding ratio) across a range of configurations, as illustrated in Figure 9. Our experimental results highlight two key observations:

- Appropriate binding effectively preserves model performance while substantially reducing both the parameter count and the computational cost of the QK circuit (scaling proportionally with the binding ratio).

- Model performance deteriorates significantly when the number of independent QK heads falls below the original MHSA head count, establishing this threshold as a critical lower bound for binding ratio selection.

## B.3 ABLATION STUDY ON QK INITIALIZATION

Given that our QK matrices maintain high dimensionality and adopt a binding strategy, a natural question arises: can we directly reuse the original MHSA QK parameters in Lorsa? To investigate this, we evaluate three settings: (1) randomly initializing the QK parameters of Lorsa, (2) initializing the QK parameters of Lorsa with the original MHSA QK parameters and allowing them to be updated during training, and (3) fixing the QK parameters to the original MHSA QK parameters throughout training. The results, summarized in Table 3, show that directly fixing the QK parameters to those of MHSA leads to worse performance compared to the other two setups. This suggests that during optimization, Lorsa learns QK parameters that capture information not present in the original MHSA parameters.

| Initialization Strategy | Fraction Variance Unexplained (FVU) |
|---|---|
| Random Initialization | 11.3% |
| Initialization with Original QK (Trainable) | 11.2% |
| Initialization with Original QK (Fixed) | 12.4% |

Table 3: Comparison of different QK initialization strategies for Lorsa.

## B.4 DOES (TOP-K) LORSA NEED RELU NON-LINEARITY TO GUARANTEE NON-NEGATIVE OUTPUTS?

To align with the superposition hypothesis and the architectural design of the SAE, we apply a ReLU to ensure that the activations $z$ are non-negative. However, we observe that this modification has negligible impact on training dynamics, as the top-$k$ activations are almost always positive for reasonable choices of $k$. This is consistent with findings reported in Gao et al. (2024).

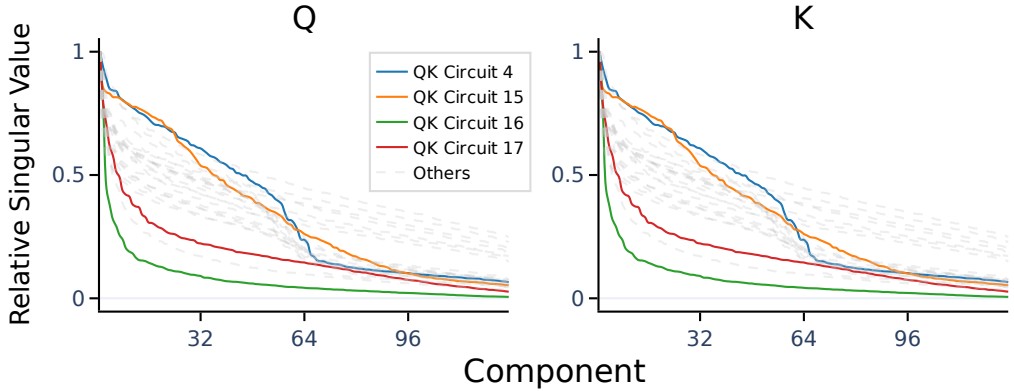

Figure 10: Sorted relative singular values of $W_Q$ and $W_K$ for each QK circuit at pythia-160m layer 5. Each circuit shows strong alignment between the spectra of $W_Q$ and $W_K$, suggesting similar structural properties. Circuits 4 and 15 have relatively high effective rank, while Circuits 16 and 17 exhibit significantly lower rank.

## C  DOES QK RANK VARY ACROSS ATTENTION UNITS?

We analyze the structure of 24 independent QK projections trained at layer 5 of Pythia-160M. Specifically, we estimate the effective rank of each pair of $W_Q$ and $W_K$ by sorting their relative singular values in descending order, as shown in Figure 10. Among these QK circuits, Circuit 4 exhibits subtoken induction, previous-token, and successor attention patterns; Circuit 15 also shows clear induction behavior. These circuits tend to have relatively high ranks. In contrast, Circuit 16 attends to itself on certain special tokens, and Circuit 17 functions as an attention sink while also attending to itself on specific inputs. Both of these circuits exhibit lower effective ranks.

## D  ADDITIONAL CASE STUDIES

### D.1  ATTRIBUTION ALGORITHM FOR IDENTIFYING LORSA HEADS WITH SPECIFIC FUNCTIONALITIES

In addition to the path patching method discussed in Section 6.1 , we employ an attribution algorithm, inspired by the approach for detecting important features with attribution in Batson et al. (2024), to identify Lorsa heads associated with specific functionalities.

The attribution score for a given Lorsa head $h$, is defined as:

$$attr_h := O_h \cdot \nabla_x \mathcal{L}$$

Here, $\nabla_x \mathcal{L}$ is the gradient of the logit on the prediction of the target token with respect to the attention output $O_h$ of the Lorsa head. For different prompt, we also try logit difference or probability difference to calculate $\nabla_x \mathcal{L}$.

quantifies the contribution of Lorsa head $h$ to the prediction of the correct token.

### D.2  EXAMPLES OF LORSA'S REDISCOVERY OF REPORTED FUNCTIONAL HEADS

The detailed information on the Lorsa heads discussed in Section 6.1 is provided in Figure 11, where we visually demonstrate the logit differences induced by the Lorsa head ,along with the most strongly correlated MSHA heads and SAE features.

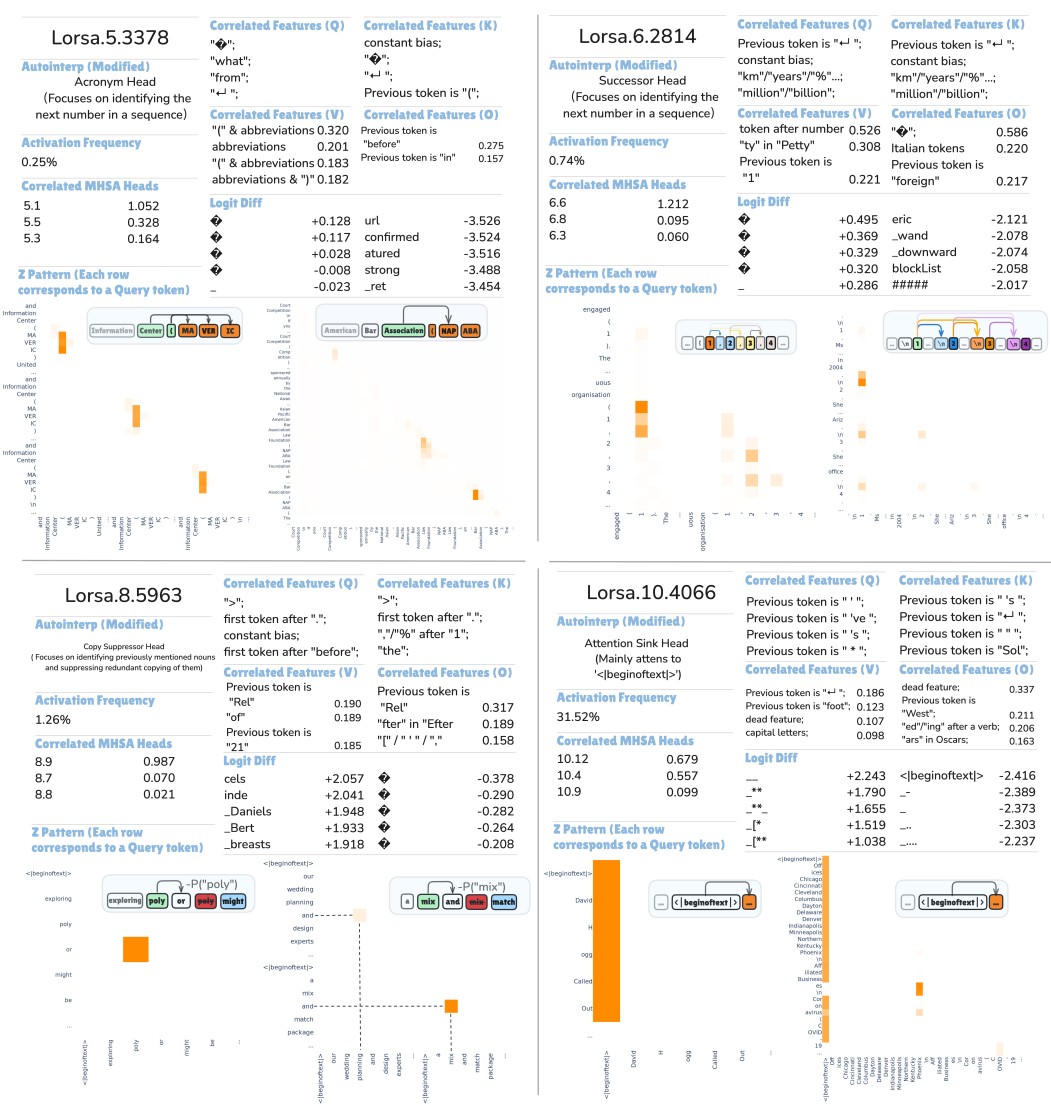

Figure 11: Detailed information on Lorsa's rediscovery of reported functional heads.

## D.3 ARITHMETIC LORSA HEADS

We present the SAE features related to the reported arithmetic Lorsa heads in Table 4, which shows consistent interpretation in terms of operand plot and $z$ pattern. Additionally, Table 5 provides a broader set of examples for these arithmetic Lorsa heads, including functional descriptions and the z-patterns of their top activations.

## D.4 PRELIMINARY PERTUBATION RESULTS

We feed Llama-3.1-8B "$75 \div 3 =$" as the clean prompt and it succeeds to predict the answer 25 ($p = 0.73$). With attribution from the correct answer logit we identify an "op2 = 3" Lorsa head in layer 15 (Lorsa.15.2668) with notable contribution. We then set the activation strength $z$ of this head to 0 at the last token position ("=") and copy its original value to a an "op2 = 5" head (Lorsa.15.3099) and rerun the forward pass from layer 15 attention. This gives an answer of 15 ($p = 0.66$).

Since $z$ of a Lorsa head indicates its output norm along the $w_o$ direction, this pertubation experiment greatly resembles steering SAE vectors (Templeton et al., 2024b). There is also an alternative

| Lorsa head ID | Manual Interpretation with Operand Plot | Manual Interpretation with $z$ Pattern |
|---|---|---|
| Lorsa.16.20791 | $\mathtt{op1} \in 27 - 43$ | near 30 |
| Lorsa.16.20931 | $\mathtt{op1} \% 10 \in [4, 5, 6]$ | ending with 4 or 6 |
| Lorsa.16.20947 | $\mathtt{op1} \% 10 \in [6, 7, 8]$ | ending with 7, sometimes 6 |
| Lorsa.15.3646 | $\mathtt{op2} \% 10 = 2$ | ending with 2 |
| Lorsa.15.3813 | $\mathtt{op2} \in 55 - 99$ | from 50 - 99 |
| Lorsa.15.4001 | $\mathtt{op2} \in 38 - 63$ | near 50 |

Table 4: Supplementary information of Lorsa Head in Figure 7. We observe alignment between interpretations obtained from operand plots and top activating $z$ patterns sampled from natural language text corpus.

| ID | Operator | Operand | Top Activation Z Pattern |
|---|---|---|---|
| Lorsa.15.3646 | Addition | op2 ends with 2 |  |
| | Subtraction | min(op1, op2) ends with 2 | |
| | Multiplication | op2 = 2 or 12 | |
| | Division | op2 = 2 | |
| Lorsa.15.3648 | Addition | op2 ends with 4 |  |
| | Subtraction | min(op1, op2) ends with 4 | |
| | Multiplication | op2 = 4, 24, or 40 | |
| | Division | op2 = 4 | |
| Lorsa.15.2668 | Addition | Inactive |  |
| | Subtraction | Inactive | |
| | Multiplication | op2 = 3, 6, 30, or 60 | |
| | Division | op2 around 3 or 30 | |
| Lorsa.15.2770 | Addition | Inactive |  |
| | Subtraction | Inactive | |
| | Multiplication | op2 around 62 and its multiples | |
| | Division | op2 around 62 and its multiples | |
| Lorsa.15.2945 | Addition | Inactive |  |
| | Subtraction | Inactive | |
| | Multiplication | op2 = 7, 11 and their multiples | |
| | Division | op2 = 7, 11 and their multiples | |

Table 5: Additional cases of arithmetic heads

interpretation that we are intervening attention computation in OV circuits - this result can be precisely achieved by swapping the $w_o$'s of these two Lorsa heads. In consequence, the pertubed Lorsa head *recieves* "$\mathtt{op2} = 3$" but *tell* subsequent computation that "$\mathtt{op2} = 5$". Such pertubation is independent from QK circuits as both Lorsa heads share the same QK weights. This serves as evidence in the wild that Lorsa heads with shared QK circuits often show similar functionalities.

### D.5 THEME ANCHOR HEADS

## E ASSESSING CORRELATION WITH MHSA

How to understand the correlation between Lorsa heads and original MHSA heads? We try to answer this by computing the attribution of each Lorsa head to the original attention heads using an oblique projection method (Appendix E.1). Analyzing all Lorsa heads trained on Pythia-160M (Appendix E.2), we find that roughly half of the Lorsa heads originate from a single original head, while the other half are superpositions across multiple original heads.

### E.1 OBLIQUE PROJECTION METHOD FOR ATTRIBUTION

Given the output of an original attention head, we project it obliquely onto the (generally non-orthogonal) basis formed by the outputs of all Lorsa heads at the same layer. The resulting coefficients

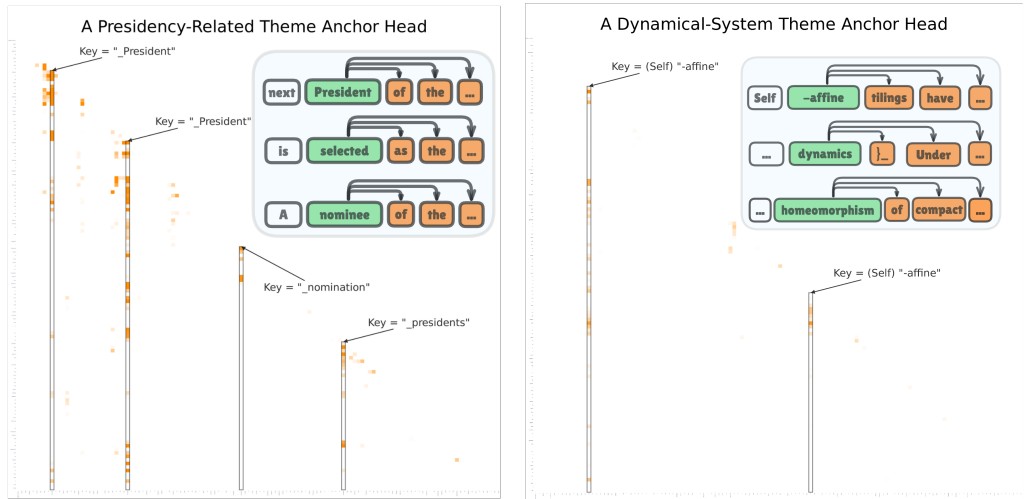

(a) $z$ pattern of a presidency-related theme anchor Lorsa head.

(b) $z$ pattern of a theme anchor Lorsa head related to dynamical systems.

Figure 12: Two examples of theme anchor Lorsa heads.

represent the contribution of the original head to each Lorsa head. Since the summed outputs of original heads and Lorsa heads closely match, the contribution coefficients for a given Lorsa head approximately sum to one. Conversely, we similarly compute the fraction of each Lorsa head's output that can be attributed to each original attention head by projecting the Lorsa head's output onto the basis formed by the original heads' outputs. All reported results are averaged over more than 1M tokens.

## E.2 How Many Attention Units are Distributed Across MHSA Heads?

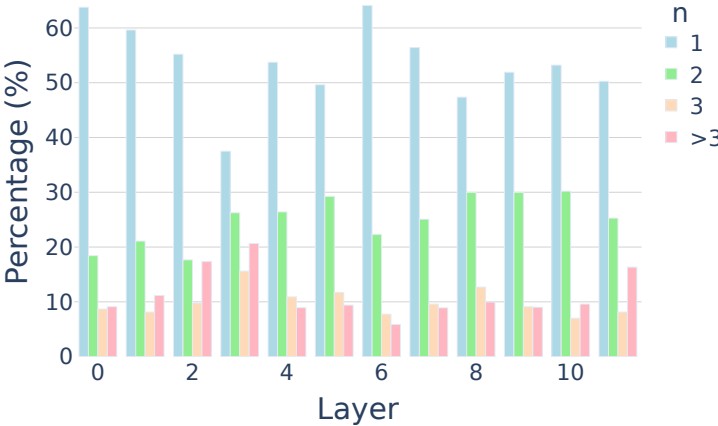

Figure 13: Distribution of Lorsa heads based on the number of original attention heads they are superposed over. No clear trend is observed across different layers. Approximately 50% Lorsa heads are primarily associated with a single original head, about 25% are superposed over two different original heads, around 10% are superposed over three different original heads, and others superposed over more than three original heads.

We compute the attribution statistics for all Lorsa heads trained on Pythia-160M. For a given Lorsa head, we define $n$ as the minimum number of original heads whose cumulative contributions exceed 90%. We interpret $n$ as the effective number of original heads a Lorsa head superposes over. As

shown in Figure 13, approximately half of the Lorsa heads are primarily derived from a single original head, about a quarter involve two original heads, and the remaining quarter involve three or more original heads.

### E.3 Induction MHSA Heads in Pythia-160M

Table 6: Contribution of each MHSA head to induction behavior in Pythia-160M, measured via path patching. Notable induction heads (`L5.0`, `L4.6`, `L5.7`, `L9.0`, `L5.6`) are bold.

| Layer\Head | 0 | 1 | 2 | 3 | 4 | 5 | 6 | 7 | 8 | 9 | 10 | 11 |
|---|---|---|---|---|---|---|---|---|---|---|---|---|
| 0 | 0.00 | 0.00 | 0.00 | 0.00 | 0.00 | 0.00 | 0.00 | 0.00 | 0.00 | 0.00 | 0.00 | 0.00 |
| 1 | 0.07 | -0.15 | -0.10 | 0.03 | 0.09 | -0.08 | -0.07 | 0.06 | -0.01 | 0.11 | 0.34 | -0.05 |
| 2 | -0.14 | 0.07 | 0.10 | 0.14 | 0.14 | -0.13 | 0.60 | -0.03 | -0.14 | 0.10 | 0.04 | 0.03 |
| 3 | -0.24 | -0.14 | -0.96 | -1.20 | -0.49 | -0.14 | 0.20 | -0.38 | -0.10 | 0.06 | -0.11 | -0.07 |
| 4 | 0.13 | -0.26 | 0.09 | -0.16 | -0.10 | -0.02 | **0.89** | 0.13 | 0.09 | -0.28 | -0.14 | 0.30 |
| 5 | **4.00** | -0.20 | 0.05 | 0.06 | -0.53 | -0.04 | **0.48** | **0.62** | 0.06 | 0.08 | 0.05 | -0.23 |
| 6 | -0.04 | -0.23 | -0.04 | -0.22 | 0.02 | 0.09 | 0.04 | -0.33 | 0.02 | -0.04 | -0.38 | 0.04 |
| 7 | -0.28 | 0.17 | 0.03 | 0.06 | -0.28 | -0.07 | 0.01 | -0.18 | -0.23 | -0.03 | -0.02 | 0.18 |
| 8 | -0.07 | 0.03 | 0.50 | 0.00 | 0.15 | -0.02 | 0.01 | -0.22 | 0.02 | -0.02 | -0.08 | 0.38 |
| 9 | **0.54** | -0.03 | 0.07 | -0.09 | -1.10 | -0.04 | 0.04 | 0.00 | 0.04 | 0.10 | -0.01 | 0.02 |
| 10 | -0.01 | 0.03 | 0.00 | 0.00 | -0.03 | -0.10 | 0.01 | -0.01 | 0.00 | -0.04 | 0.03 | 0.01 |
| 11 | -0.14 | -0.13 | -0.05 | -0.04 | 0.00 | -0.02 | -0.11 | -0.02 | 0.01 | -0.07 | -0.02 | 0.06 |

We use path patching to measure the contribution of each MHSA head in Pythia-160M to induction behavior. The results are shown in Table 6. We find that heads `L5.0`, `L4.6`, `L5.7`, `L9.0`, `L5.6` exhibit the most prominent induction signals.

## F   Interaction Between Lorsa Heads and SAE Features

We trained Sparse Autoencoders (SAE) on both the inputs and outputs of Lorsa to facilitate the understanding of its functionality. Since Lorsa's Q, K, and V are computed from the input, with the output derived from O contributing to the final result, interactions between SAE features and these components exist across all four aspects: Q, K, O, and V. To evaluate the influence of SAE features on Q and K, we employ an ablation method (Appendix F.1). The correlation between the $OV$ and SAE features is assessed using cosine similarity (Appendix F.2). For each Lorsa head, we identify the SAE features most strongly correlated with different aspects. The results are visualized in the Lorsa head dashboard.

### F.1   Quantifying Feature Impacts on Q and K

For a given Lorsa head, the impact of a specific feature on Q is calculated as follows: First, we compute the attention pattern at the activation locations of the Lorsa head. Then, the feature is ablated from the input, and $Q'$ and the new attention pattern are computed (with K remaining unaffected). The Kullback-Leibler (KL) divergence between the original and modified attention patterns is used to quantify the effect of the feature on Q. After iterating over 1 million tokens, the maximum KL divergence observed across all activations of the Lorsa head is taken as the measure of the feature's influence on Q for this head. A similar approach is used to calculate the impact of a feature on K, with the difference being that when recalculating the attention pattern, all instances of K are recomputed using the modified input, while Q remains unchanged.

### F.2   Quantifying Direct Feature Attribution via O and V

For a given Lorsa head, both the weight vectors $W_O$ and $W_V$ are one-dimensional vectors of size $D_{\text{model}}$. Therefore, for each SAE feature trained on the Lorsa input, the contribution to $V$ is linear, meaning that the contribution of each feature to $V$ scales proportionally with the feature's activation value. Similarly, for each activation $z$ of the head, the contribution of SAE features trained on the Lorsa output to the activation value is also linear. We compute the cosine similarity between the decoder of each SAE feature trained on the Lorsa input and $W_V$, which quantifies its correlation

with $V$ for the given Lorsa head. Similarly, the cosine similarity between the encoder of each SAE feature trained on the Lorsa output and $W_O$ is computed to measure its correlation with $O$ for the given Lorsa head.

## G  LORSA DARK MATTER

Figure 14 illustrates the per-token error norms of Lorsa and SAE across layers 2, 6, and 10 of Pythia-160M on a set of 64 tokens. Figure 15 quantifies the distribution of cosine similarity between Lorsa and SAE's per-token error norms on the same layers, measured on approximately 10,000 tokens. These results indicate that the loss pattern between pre token between Lorsa and SAE has a nontrivial correlation.

It is interesting that both Lorsa and SAE exhibit a positive correlation in their magnitudes and trends for FVU and per-token error norms.

We propose that this is not a coincidence, and hypothesize that it stems from a shared gap between sparse dictionary learning and the representation structure of data within the model. Alternatively, this correlation may arise from the challenge that sparse dictionary learning faces in capturing super-rare data features or certain nonlinear or dense components within the features.

This supports the hypothesis of *universal dark matters* (Olah et al., 2020; Engels et al., 2024) that a certain fraction of error results from the superposition hypothesis itself that cannot be addressed simply with larger Lorsas (SAEs).

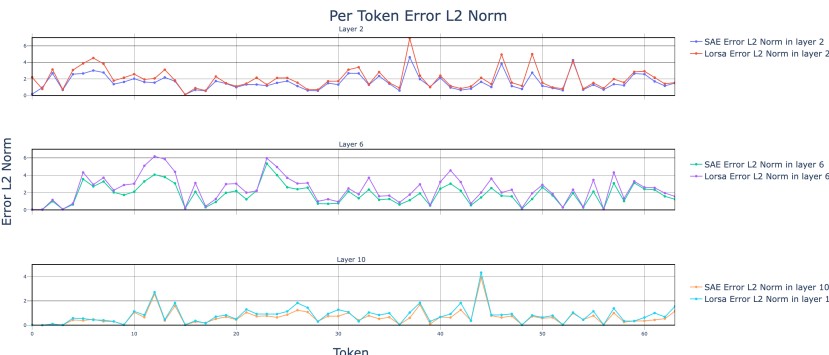

Figure 14: Per-token error norms of Lorsa and SAE on layer 2, 6, and 10 of Pythia-160M for a randomly sampled sequence with 64 tokens.

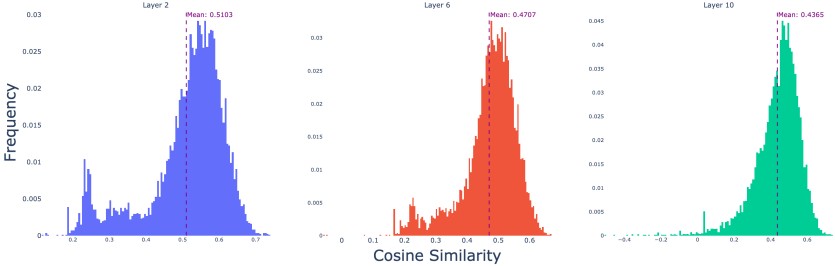

Figure 15: Cosine similarity distribution of per-token error between Lorsa and SAE on layer 2, 6, and 10 in Pythia-160M, measured with approximately 10,000 tokens.

## H TOWARDS FULL SPARSIFICATION OF A 2-LAYER TRANSFORMER

Since our final goal is to understand Transformers' inner working by breaking down MHSA and MLPs into atomic units (Figure 1), we train Lorsa and Transcoder (Dunefsky et al., 2024) on a 2-layer Transfomer (link). We follow the method introduced in Ge et al. (2024) where they multiply features via QK circuit to find the most salient feature pairs contributing to QK scores. Alternatively applying attribution through Transcoder features / Lorsa heads and QK ablation gives us the clear attribution graph for induction behavior (Figure 16). Due to the capability constraint of this model, we failed to observe more interesting behaviors or attribution graphs involving Transcoder features. Nonetheless, we believe applying Lorsa and Cross-Layer Transcoders (Ameisen et al., 2025) to a larger model may reveal a lot of surprising behaviors, following the spirit of Lindsey et al. (2025).

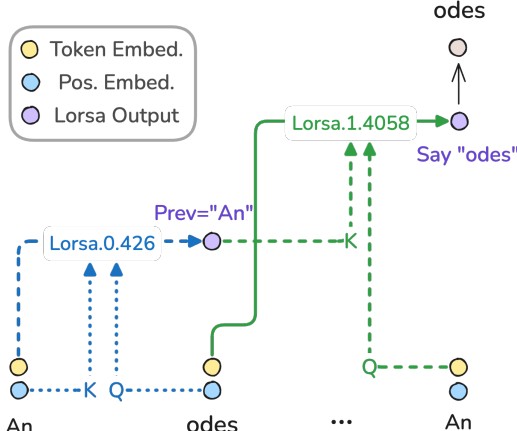

Figure 16: An induction circuit found in our fully sparsified replacement model.

## I AUTOMATED INTERPRETABILITY DETAILS

**Evaluation Protocol.** Our automated interpretability assessment employs a two-phase explanation-simulation paradigm adapted from Bills et al. (2023):

1. **Explanation Phase**: GPT-4o generates mechanistic explanations using:
   - For activation patterns: 8 top-activating token contexts
   - For $z$-patterns/DFAs: Contribution graphs to max-activating tokens
2. **Simulation Phase**: GPT-4o predicts activations/patterns for:
   - 4 top-activating contexts (testing pattern recognition)
   - 4 randomly sampled contexts (testing generalization)

**Top Activation Explanation Phase Prompt.**

---

**Prompt**

We are analyzing the activation levels of features in a neural network, where each feature activates certain tokens in a text. Each token's activation value indicates its relevance to the feature, with higher values showing stronger association. Your task is to infer the common characteristic that these tokens collectively suggest based on their activation values.
Consider the following activations for a feature in the neural network. Activation values are non-negative, with higher values indicating a stronger connection between the token and the feature. Summarize in a single sentence what characteristic the feature is identifying in the text.

---

> Doń list examples of words. Do not start with "This feature is identifying...". Go straight to the explanation.
> Sentence 1:
> <START>
> <|endoftext|><tab>-0.0
> /<tab>-0.0
> */<tab>0.2
> ...(omitted)
> <END>
> Sentence 2:
> ...(omitted)

**Top Activation Simulation Phase Prompt.**

---

**Prompt**

We're studying neurons in a neural network. Each neuron looks for certain things in a short document. Your task is to read the explanation of what the neuron does, and predict the neuron's activations for each token in the document.

For each document, you will see the full text of the document, then the tokens in the document with the activation left blank. You will print the exact same tokens verbatim, but with the activation values filled in according to the explanation. Pay special attention to the explanation's description of the context and order of tokens or words.

Fill out the activation values with integer values from 0 to 10. Don't use negative numbers. Please think carefully. No need to include rationales. Directly start with the first token and do not use code blocks, i.e., "'.

Neuron 1 explanation: This feature is indentifying vowels.
Sequence 1: Tokens without Activations:
a<tab>
b<tab>
c<tab>
d<tab>
e<tab>
f<tab>
Sequence 1 Tokens with Activations:
a<tab>10
b<tab>0
c<tab>0
d<tab>0
e<tab>10
f<tab>0
Neuron 2 explanation: <Autointerp explanations generated in the previous phase>
<Few shot examples>

---

*z* **Pattern / DFA Explanation Phase Prompt.**

---

**Prompt**

We are analyzing the attention map of attention heads in a neural network, where each head attends between tokens in a text. Given a head and a query token, we provide each previous tokeń contribution value, with higher values showing stronger association. Your task is to infer the common characteristic of this head that these sequences collectively suggest based on their attention map.

Consider the following attention maps for an attention head. Each line is in the format of <token><tab><value>. Query tokens are additionally highlighted with <to-

---

> ken><tab><value><tab>**Query token**. Note that query tokens also attend to themselves.
> Higher values indicates a stronger contribution from this token to the query token.
> Summarize in a single sentence what characteristic the head is attending from and to in the text.
> It might be helpful to summarize both the commonality of query tokens and source tokens (if
> any). It is also recommended to mention if this head is often attending to itself.
> Don't list examples of words. Do not start with "This head is ...". Directly start with the
> explanation.
> Sentence 1:
> <START>
> <|endoftext|><tab>-0.0
> /<tab>0.0
> ...(omitted)
> */<tab>0.0<tab>**Query token**

### $z$ Pattern / DFA Simulation Phase Prompt.

> **Prompt**
>
> We're studying attention heads in a neural network. Each head follows a certain attention pattern
> in a short document. Your task is to read the explanation of what the head does, and predict the
> head's attention pattern for each previous token in the document, given a specific query token.
> For each document, you will see the full text of the document, then the tokens in the document
> with the activation left blank. You will print the exact same tokens verbatim, but with the contri-
> bution values filled in according to the explanation. Pay special attention to the explanation's
> description of the context and order of tokens or words.
> Each line is in the format of <token><tab>. Query tokens are additionally highlighted with
> <token><tab>**Query token**<tab>.
> Fill out the contribution values with integer values from 0 to 10. Don't use negative numbers.
> Please think carefully. No need to include rationales. Directly start with the first token and do
> not use code blocks, i.e., "'.
> Head 1 explanation: This head is attending from one vowel to previous vowels and itself.
> Sequence 1 Tokens without Activations:
> a<tab>
> b<tab>
> c<tab>
> d<tab>
> e<tab>**Query token**
> Sequence 1 Tokens with Activations:
> a<tab>10
> b<tab>0
> c<tab>0
> d<tab>0
> e<tab>**Query token**<tab>10
> Head 2 explanation: <Autointerp explanations generated in the previous phase>
> <Few shot examples>

## J THE PATCHING ATTRIBUTION APPROXIMATION BOUND BETWEEN MHSA AND LORSA

**Definition 1** (MHSA). *For attention module, the calculation by MHSA can be formalized as*

$$\mathscr{A}_{MHSA}(\mathbf{x}) = h_1(\mathbf{x}) + \cdots + h_n(\mathbf{x}), \tag{1}$$

*where $n$ is the number of attention heads, $\mathbf{x} \in \mathbb{R}^d$ is the input, and $h_i$ is the $i$-th attention head.*

**Definition 2** (Lorsa). *For attention module, the calculation by Lorsa can be formalized as*

$$\mathscr{A}_{Lorsa}(\mathbf{x}) = \sum_{j=1}^{N} TopK(p_j(\mathbf{x}))\hat{h}_j(\mathbf{x}), \tag{2}$$

*where $N >> n$ is the number of Lorsa heads, $\hat{h}_j$ is the $j$-th Lorsa head defined in previous Section ..., $p_j : \mathbf{x} \mapsto \mathbb{R}$ represent the activation of $h_j$, and TopK is the TopK activation. Specifically, the TopK activation function can be expressed by*

$$TopK(p_j(\mathbf{x})) = \begin{cases} p_j(\mathbf{x}), & p_j(\mathbf{x}) \text{ is in the top-k activations,} \\ 0, & p_j(\mathbf{x}) \text{ is not in the top-k activations,} \end{cases} \tag{3}$$

From the Linear Representation Hypothesis, we assume that the attention head in MHSA can be approximated by the the linear combination of Lorsa Heads, and the approximation error is bounded:

**Assumption 1** (Linear Representation Hypothesis of Attention). *For each MHSA attention head $h_i$, there exists a Lorsa head set $\mathbf{S}_i$ satisfying*

$$\mathscr{A}_{MHSA}(\mathbf{x}) = \sum_{j\in\mathbf{S}_i} p_j(\mathbf{x})\hat{h}_j(\mathbf{x}) + \epsilon_i(\mathbf{x}), \tag{4}$$

*where $\epsilon_i(\mathbf{x}) > 0$ is the approximation error from Lorsa. The approximation between MHSA and Lorsa is bounded, i.e., there exists $\epsilon > 0$ satisfying*

$$\|\mathscr{A}_{MHSA}(\mathbf{x}) - \mathscr{A}_{Lorsa}(\mathbf{x})\| \le \epsilon, \ \forall \mathbf{x} \in \mathbf{D}, \tag{5}$$

*where $\mathbf{D}$ is the dataset.*

Previous studies have also referred to this estimation error as dark matter, which is inevitable.

Moreover, from the superposition hypothesis, the activation of Lorsa heads is sparse for each input. And, since we initialize Lorsa's QK module by MHSA, it is natural to assume that the Lorsa head will align with the head of a specific MHSA. Therefore, we have the below assumption.

**Assumption 2** (Superposition Hypothesis). *For Lorsa, the activation is sparse, i.e., for any Lorsa head set $\mathbf{S}$, we have*

$$\sum_{j\in\mathbf{S}} notTopK(p_j(\mathbf{x}))\hat{h}_j(\mathbf{x}) \approx \mathbf{0}, \tag{6}$$

*where notTopK is defined similar to TopK in eq. 3.*

*For the MHSA attention head, we have the Lorsa heads sets $\{\mathbf{S}_i\}$ in eq. 4 for each MHSA head is a partition of the all Lorsa heads, i.e.,*

$$\begin{aligned} S_i \cap S_j &= \emptyset, \text{ for } i \ne j, \\ \bigcup S_i &= \{1, 2, \cdots, N\}. \end{aligned} \tag{7}$$

Therefore, we can prove that, from the perspective of patching, the behavior of the $i$-th MHSA attention head is approximately equivalent to that of the Lorsa head in $\mathbf{S}_i$, i.e., this sparsification does not alter the model's underlying behavior in feature-level. First, following the direct logit attribution (DLA) (Wang et al., 2022), we define the influence of the heads in MHSA and Lorsa.

**Definition 3** (Variation for DLA in MHSA and Lorsa). *The variation for DLA (VDLA) of $i$-th MHSA heads for the input pair $(\mathbf{x}_r, \mathbf{x}_c)$ ($\mathbf{x}_r$ is the reference input, and the $\mathbf{x}_c$ is the counterfactual input transformed from $\mathbf{x}_r$) can be defined as*

$$VDLA_{MHSA}(\mathbf{x}_r, \mathbf{x}_c, i) := f(h_i(\mathbf{x}_r)) - f(h_i(\mathbf{x}_c)), \tag{8}$$

*where $f : \mathbb{R}^d \to \mathbb{R}$ is the composite map for DLA. And we assume that the $f$ is Lipschitz continuous, i.e., there exists Lipschitz bound $C > 0$ such that*

$$|f(\mathbf{x}) - f(\mathbf{y})| \le C|\mathbf{x} - \mathbf{y}|. \tag{9}$$

*And the VDLA of Lorsa head sets $\mathbf{S}$ for the input pair $(\mathbf{x}_r, \mathbf{x}_c)$ can be defined as*

$$VDLA_{Lorsa}(\mathbf{x}_r, \mathbf{x}_c, \mathbf{S}) := f\left(\sum_{j\in\mathbf{S}} TopK(p_j(\mathbf{x}_r))\hat{h}_j(\mathbf{x}_r)\right) - f\left(\sum_{j\in\mathbf{S}} TopK(p_j(\mathbf{x}_c))\hat{h}_j(\mathbf{x}_c)\right). \tag{10}$$

The VDLA metric reflects the strength of influence exerted by certain heads in MHSA and Lorsa on model behavior. And we can prove that the influences in MHSA and Lorsa are approximately equivalent by the theorem below.

**Theorem 1** (The VDLA Approximation Bound between MHSA and Lorsa). *From the Assumption 1 and 2, we have*

$$|VDLA_{MHSA}(\mathbf{x}_r, \mathbf{x}_c, i) - VDLA_{Lorsa}(\mathbf{x}_r, \mathbf{x}_c, \mathbf{S}_i)| \lesssim 2C\epsilon, \tag{11}$$

*where $\epsilon$ is the error bound defined in Assumption 1, and $C$ is the lipschitz bound of $f$ defined in Definition 3.*

*Proof.* For the VDLA error, we have

$$|VDLA_{MHSA}(\mathbf{x}_r, \mathbf{x}_c, i) - VDLA_{Lorsa}(\mathbf{x}_r, \mathbf{x}_c, \mathbf{S}_i)|$$

$$\leq \left| f(h_i(\mathbf{x}_r)) - f(h_i(\mathbf{x}_c)) - f\left(\sum_{j\in\mathbf{S}} \text{TopK}(p_j(\mathbf{x}_r))\hat{h}_j(\mathbf{x}_r)\right) + f\left(\sum_{j\in\mathbf{S}} \text{TopK}(p_j(\mathbf{x}_c))\hat{h}_j(\mathbf{x}_c)\right) \right|$$

$$\leq \left| f(h_i(\mathbf{x}_r)) - f\left(\sum_{j\in\mathbf{S}} \text{TopK}(p_j(\mathbf{x}_r))\hat{h}_j(\mathbf{x}_r)\right) \right| + \left| f(h_i(\mathbf{x}_c)) - f\left(\sum_{j\in\mathbf{S}} \text{TopK}(p_j(\mathbf{x}_c))\hat{h}_j(\mathbf{x}_c)\right) \right|$$

$$\leq C \left| h_i(\mathbf{x}_r) - \sum_{j\in\mathbf{S}} \text{TopK}(p_j(\mathbf{x}_r))\hat{h}_j(\mathbf{x}_r) \right| + C \left| h_i(\mathbf{x}_c) - \sum_{j\in\mathbf{S}} \text{TopK}(p_j(\mathbf{x}_c))\hat{h}_j(\mathbf{x}_c) \right| \tag{12}$$

From the Assumption 1, for the first term, we have

$$\left| h_i(\mathbf{x}_r) - \sum_{j\in\mathbf{S}} \text{TopK}(p_j(\mathbf{x}_r))\hat{h}_j(\mathbf{x}_r) \right|$$

$$= \left| \epsilon_i(\mathbf{x}_r) + \sum_{j\in\mathbf{S}} \text{notTopK}(p_j(\mathbf{x}_r))\hat{h}_j(\mathbf{x}_r) \right| \tag{13}$$

$$\leq |\epsilon_i(\mathbf{x}_r)| + \left| \sum_{j\in\mathbf{S}} \text{notTopK}(p_j(\mathbf{x}_r))\hat{h}_j(\mathbf{x}_r) \right|.$$

From the Assumption 2, for all $\mathbf{x} \in \mathbf{D}$, we have

$$\|\mathscr{A}_{MHSA}(\mathbf{x}) - \mathscr{A}_{Lorsa}(\mathbf{x})\| = \|\sum_{j=1}^{n} \epsilon_j(\mathbf{x})\| = \sum_{j=1}^{n} \epsilon_j(\mathbf{x}) \leq \epsilon, \tag{14}$$

where the second equality is from $\epsilon_i(\mathbf{x}) > 0$. Therefore, we have

$$\epsilon_j(\mathbf{x}) \leq \epsilon. \tag{15}$$

From the eq. 13 and eq. 15, we have

$$\left| h_i(\mathbf{x}_r) - \sum_{j\in\mathbf{S}} \text{TopK}(p_j(\mathbf{x}_r))\hat{h}_j(\mathbf{x}_r) \right|$$

$$\leq \epsilon + \left| \sum_{j\in\mathbf{S}} \text{notTopK}(p_j(\mathbf{x}_r))\hat{h}_j(\mathbf{x}_r) \right|. \tag{16}$$

Then, from the Assumption 2, we have

$$\left| h_i(\mathbf{x}_r) - \sum_{j\in\mathbf{S}} \text{TopK}(p_j(\mathbf{x}_r))\hat{h}_j(\mathbf{x}_r) \right| \lesssim \epsilon. \tag{17}$$

Similarly, for the second term in eq. 12, we have

$$\left| h_i(\mathbf{x}_c) - \sum_{j \in \mathbf{S}} \text{TopK}(p_j(\mathbf{x}_r)) \hat{h}_j(\mathbf{x}_c) \right| \lesssim \epsilon. \tag{18}$$

Substituting eq. 17 and eq. 18 into eq. 12, we have

$$|VDLA_{MHSA}(\mathbf{x}_r, \mathbf{x}_c, i) - VDLA_{Lorsa}(\mathbf{x}_r, \mathbf{x}_c, \mathbf{S}_i)| \lesssim 2C\epsilon. \tag{19}$$

The proof is completed. $\qquad\square$

From the Theorem 1, we obtain the following corollary.

**Corollary 1.** *For the dataset $\mathbf{D}_r$, where $(\mathbf{x}_r, \mathbf{x}_c) \sim \mathbf{D}_r$, $\mathbf{x}_r \sim \mathbf{D}$ is the reference input, and $\mathbf{x}_c$ is counterfactual input transformed from $\mathbf{x}_r$, $\mathbf{D}$ is the original input dataset, we have*

$$\mathbb{E}_{(\mathbf{x}_r, \mathbf{x}_c) \sim \mathbf{D}_r} |VDLA_{MHSA}(\mathbf{x}_r, \mathbf{x}_c, i) - VDLA_{Lorsa}(\mathbf{x}_r, \mathbf{x}_c, \mathbf{S}_i)| \lesssim 2C\epsilon \tag{20}$$

