# OpenReview forum: "Towards Understanding the Nature of Attention with Low-Rank Sparse Decomposition"
_ICLR.cc/2026/Conference — ICLR 2026 Poster_

### Official Review · Reviewer_8qJU · 2025-10-30

**Soundness:** 2
**Presentation:** 3
**Contribution:** 3
**Rating:** 4
**Confidence:** 2

**Summary:**

This paper studies attention superposition by introducing **Low-Rank Sparse Attention (Lorsa)**—a per-layer module trained to **reconstruct** an MHSA layer’s output as a top-$K$ sum of many rank-1 OV “attention units,” with shared QK parameters to control cost. Training minimizes an MSE objective against the layer’s MHSA output (Algorithm 1), yielding a sparse, overcomplete basis over the attention-output space that can be analyzed with standard interpretability tooling. The authors apply Lorsa to Pythia-160M and Llama-3.1-8B, report fidelity–sparsity scaling and per-layer FVU, and present qualitative/automated evidence that Lorsa rediscovers known head families (induction, successor, sinks) and surfaces new phenomena (subtoken-induction and arithmetic-specific heads), while broadly tracking SAE trends in reconstruction and interpretability metrics. Lorsa is compatible with RoPE and GQA through straightforward adaptations.

**Strengths:**

First, the methodological design is original yet well-justified: the rank-1 OV constraint sharpens semantics; shared-QK binding preserves QK dimensionality where needed and avoids the degradation observed when QK is reduced below the MHSA head dimension. The top-$K$ sparsity yields monosemantic components that are easy to inspect. The paper provides clear algorithms and ablations supporting these choices.

Second, the presentation empirical analysis is substantive and yields new insights. Lorsa tracks SAE scaling and per-layer patterns, recovers canonical heads, and identifies novel mechanisms (e.g., arithmetic-specific and subtoken-induction heads) in Llama-3.1-8B, with preliminary causal probes that connect heads to behavior.

**Weaknesses:**

- My main concern is with the paper’s positioning: the manuscript repeatedly states that “Lorsa serves as a replacement model for Transformer attention”. However, the scope of evaluation is strictly __per-layer reconstruction__ and interpretability; there is no end-to-end evaluation or deployment-oriented analysis. As written, the “replacement” phrasing risks misleading readers about the paper’s primary contribution, which is pattern understanding of MHSA via decomposition.
    - A related, practical limitation is the scale of the Llama-3.1-8B setting: ~32K Lorsa heads per layer with (K=128) and ~512M parameters per layer (vs. 64M for MHSA) make the module expensive, and training (more like reconstructing for approximation) each Llama layer’s Lorsa takes ~24 A100-hours under the stated batch size. This underscores that the artifact is currently best viewed as an interpretability tool rather than a deployable substitute .
- A second concern is _behavioral equivalence vs. output approximation_. I am not totally convinced that closely approximating MHSA outputs implies similar underlying behavior. For instance, two architectures (A) and (B) can be trained to produce near-identical outputs while differing substantially in inductive biases and internal mechanisms. Although Algorithm 1 shows Lorsa as a simplified Top-K variant of MHSA, architectural changes e.g. rank-1 OV, shared-QK, and Top-(K) gating may alter how features are represented and composed. Thus, even if $||\mathrm{Lorsa}(x)-\mathrm{MHSA}(x)||$ is small, approximation alone does not justify studying them “as the same”; a theoretical link (e.g., bounds ensuring stability of feature-level attributions or circuit-level predicates) is needed, particularly given superposition and shared-QK.

**Questions:**

- On theoretical grounding, can you provide a result formalizing when Lorsa is a faithful surrogate for studying MHSA? For instance, assume a layerwise error bound $||\text{Lorsa}(x)-\text{MHSA}(x)||\le \varepsilon$ and conditions on QK similarity; then prove stability of **linear readouts** and **attribution functionals** (e.g., path-patching scores, projection-based attributions) to errors of size $\varepsilon$. A theorem of the form
   $$
   \mathbb{E}_{x\sim \mathcal{D}}|| \mathcal{A}(\text{Lorsa};x) - \mathcal{A}(\text{MHSA};x) ||_p \le C\cdot \varepsilon,
   $$for a relevant class $\mathcal{A}$ would directly justify that we can study MHSA via Lorsa. Clarifying how shared-QK and Top-K selection affect such stability (e.g., via Lipschitz or margin assumptions on head activations $z$) would be especially valuable.
- I am not an expert in this field, so Id like to refer to other reviewers for the final recommendation.

---

> ### Author Response · Authors · 2025-11-23
> **Positioning of Lorsa and the term of "replacement models"**
>
> We sincerely thank you for your suggestions. Your concerns and questions actually touch the heart of many important problems in mech interp.
>
> > On the paper’s positioning. The “replacement” phrasing risks misleading readers about the paper’s primary contribution; A related, practical limitation is the scale of the Llama-3.1-8B setting.
>
> Thank you for pointing this out. The term of "replacement models" was first introduced in [Anthropic's Circuit Tracing paper](https://transformer-circuits.pub/2025/attribution-graphs/methods.html#building) (as far as we know). In this paper, the authors systematically studied substituting all MLP layers with their replacement model end to end and found that model performance significantly degraded. We have not included such analysis due to space and computation resource constraint, and we expect to also see similar results if we apply Lorsa to all layers. It will be severer if we simutaneously replace attention and MLP, i.e. fully sparsifying a transformer.
>
> This may indeed lead to confusion, especially for non-interp readers. Our intended positioning of this paper is to introduce a sparse, interpretable replacement model for attention, complementary to MLP replacement models (i.e. Transcoders) so that we have the basic techniques needed for fully sparsification. We have now clarified this in "Note on terminology" in the intro section in our modified manuscript.
>
> The overcomplete nature in the whole SAE literature is a central problem. We can leverage some sparse kernels to accelerate training or inference, but in general this overparameterization would pose challenges to deployment. Indeed, replacement models are currently an interpretability tool to generate hypothesis of transformer mechanisms, rather than a deployment tool to simultaneously achieve efficiency and interpretability.

---

> ### Author Response · Authors · 2025-11-23
> **An attempt to bound mechanistic faithfulness of replacement models and a counterexample**
>
> We thank you for your suggestions and questions on mechanistic faithfulness of sparse replacement models. We give a required form of proof in Appendix J of the revised manuscript. We did not show it here due to OpenReview formatting issues. The proof sketch is as follows:
>
> Metric (Variation of Direct Logit Attribution, VDLA):
>
> Measures the influence of a head (or head-set) by the change in model output when replacing reference input with counterfactual input. The function \\( f \\) is Lipschitz continuous with constant \\( C \\).
>
> Proof sketch:
>    - The difference in VDLA between MHSA and Lorsa is bounded using the triangle inequality and Lipschitz property.
>    - The approximation error from MHSA to Lorsa is bounded by ε (from Linear Representation Hypothesis).
>    - The sparse activation error is negligible (from Superposition Hypothesis).
>    - Combining these gives the final bound: \\( \\lesssim 2C\\epsilon \\).
>
>
> Despite the bound we give above, it is still possible that replacement models are not strictly mechanistically faithful to the underlying module. A counterexample and discussions of potential solutions is given in Anthropic's recent post of [Toy Model of Mechanistic (Un)Faithfulness](https://transformer-circuits.pub/2025/faithfulness-toy-model/index.html). We believe there is much space for sanity checks and improvement in the whole replacement model paradigm.
>
> We thank you again for your insightful questions. And we are looking forward to further discussions.

---

### Official Review · Reviewer_g9Wx · 2025-10-31

**Soundness:** 4
**Presentation:** 2
**Contribution:** 3
**Rating:** 8
**Confidence:** 4

**Summary:**

This paper introduces Low-Rank Sparse Attention (Lorsa), a sparse replacement model for MHSA designed to tackle the problem of attention superposition. To do that, the authors decompose the OV circuit into a large overcomplete set of sparse rank-1 heads. The work demonstrates that this method successfully rediscovers known mechanisms (like induction heads and successor heads) and uncovers novel, finer-grained units (such as subtoken induction heads and specific arithmetic heads). The paper provides quantitative analysis showing that the interpretability of Lorsa heads is comparable to that of Sparse Autoencoder (SAE) features, positioning Lorsa as a promising new tool for the mechanistic interpretability of attention layers.

**Strengths:**

- The paper addresses attention superposition, a critical but less-studied counterpart to feature superposition in MLPs. The application of sparse dictionary learning to decompose attention head outputs (specifically via rank-1 OV circuits) is a novel and significant methodological contribution, providing a much-needed alternative to SAEs for interpreting attention.
- The method is well-validated. The authors successfully recover a wide range of previously identified attention mechanisms, which serve as a strong sanity check. More importantly, the discovery of novel, fine-grained units like "subtoken induction heads" and families of "arithmetic-specific Lorsa heads" demonstrates the utility and resolving power of the method.
- The authors provide a strong quantitative benchmark by comparing Lorsa's interpretability directly against SAEs using automated interpretability methods. The paper also includes valuable ablation studies (in the appendix) that explore the architectural design choices, such as the impact of QK dimension and initialization.

**Weaknesses:**

- The notation is confusing:
    - The total number of Lorsa heads is denoted as $H$ in Algorithm 1 (line 140) but as $N_{Lorsa}$ in the main text (e.g., line 185). This is compounded in Section 4.1, where $N$ is used for the number of parameters.
    - The term $D_{QK}^{Lorsa}$ is used in Section 3.1 (line 169) without any prior definition. Its usage is also conflated: it appears to mean a dimension in line 169 ("...as $D_{QK}^{Lorsa}$ decreases") but a count of heads in line 171 and the Table 1 caption ("...across every $D_{QK}^{Lorsa}$ heads"). This makes the QK sharing mechanism difficult to parse.
- There is a direct contradiction between the pseudocode and the main text describing the model's architecture:
    - Algorithm 1 explicitly shows per-head query and key weights ($W_q^h, W_k^h$), implying no parameter sharing.
    - Section 3.1 (lines 168-183) explicitly describes the opposite: a parameter-sharing mechanism where groups of Lorsa heads share a single QK circuit. This made the paper unclear while reading, as the QK sharing mechanism is a core part of the Lorsa design.
- Some important limitations and conceptual discussions are in the Appendix. I understand that this could be because of spacing issues, so I would definitely recommend having these discussions in the camera-ready version of the paper.

**Questions:**

**Questions:**

1. Could the authors please clarify the difference between Algorithm 1 and the text description?
2. Could you please clarify the notation issues mentioned in the Weakness section?
3. The FVU values, while comparable to SAEs, are non-trivial. Appendix H introduces the Lorsa Dark Matter hypothesis, suggesting this error is structured and shared with SAEs. This is a critical point for the entire replacement model paradigm. Could the authors elaborate on the implications of this? If 10-30% of the variance is unexplained and potentially uninterpretable by replacement models based methods, how does this limit the functional completeness of the discovered units and the conclusions we can draw from them?


**Minor comments:**
There is a typo in line 126: “privleged”

---

> ### Author Response · Authors · 2025-11-23
> **Clarity Improvements & Discussions on "dark matters"**
>
> We are very grateful for your recognition and pointing out several important clarity issues. And we have included the discussion / limitation section in the main text. Your advice greatly helps improve the clarity of this manuscript.
>
> > contradiction between Algorithm 1 and the text description
>
> We have clarified in the second line of Algorithm 1 that these heads are not indepedently indexed by $h$ and are under some weight binding constraint.
>
> > notation issues in the number of Lorsa heads and the dimension of Lorsa QK circuits.
>
> We use a more consistent notation of $H_{Lorsa}$ to denote the number of Lorsa heads, avoiding confusion with $N$ in scaling law contexts.
>
> We use $G$ to denote the number of Lorsa heads sharing one QK circuit, and state that by default we adopt $G=D^{QK}_{Lorsa}$.
>
> All modifications above are now included in the revised manuscript.
>
> > On the Dark Matter hypothesis
>
> The study of irreducible error in SAE scaling laws (i.e. dark matter) has been an open question in the field. [Engels et al.](https://arxiv.org/abs/2410.14670) found that a proportion of irreducible error can be predicted linearly from the input, but they failed to systematically explain its origin. Nonetheless, their explanatation that this might be dense linear features, and the residual was "Nonlinear error" makes intuitive sense.
>
> We have some very preliminary explorative experiments showing sparse dictionary learning methods, even trained on activations composed solely with linear, sparse features, fail to learn the true underlying features with small activation values or frequency. This might be due to the negligible supervision signal these features provided during training. [A related solution](https://aclanthology.org/2025.findings-naacl.87/) is training on a narrow distribution of data we are particularly interested in.
>
> Since language models themselves are not perfect, their activation might not be as well. For instance, attention heads may attend to totally unrelated tokens for no reason. And embeddings of rare tokens might be pure noise, etc. To make things worse, this noise will accumulate through layers and token positions. We conjecture this may be the main source of nonlinear error.
>
> We are currently unable to go further in this direction. But we think at the end of the day, it might not be necessary to fully explain activations to fully understand the model's behavior, as long as we are able to identify which part of dark matter does not require explaining.
>
> From a pragmatical perspective, there is still large space for improvement in Lorsa (as well as transcoders) architecture, initialization and training strategy to further push the scaling law of these replacement models so we can see more interesting behaviors of the model by simply scaling up these replacement models. But eventually the community will have to come back to the dark matter question.
>
> We thank you again for your valuable advice and questions. And we are looking forward to further discussions.

---

> > ### Comment · Reviewer_g9Wx · 2025-11-27
> >
> > Thanks for all the clarifications. Indeed the fix of the clarity issues and notation improve the readability of the paper. I will keep my score.

---

> ### Author Response · Authors · 2025-11-27
>
> We thank you again for your valuable comments and suggestions.

---

### Official Review · Reviewer_ju5a · 2025-11-01

**Soundness:** 3
**Presentation:** 3
**Contribution:** 3
**Rating:** 6
**Confidence:** 2

**Summary:**

This paper introduces Low-Rank Sparse Attention (Lorsa), a method to decompose transformer attention into sparser, more interpretable components. It effectively addresses "attention superposition," rediscovering known mechanisms like induction heads and identifying novel ones (e.g., arithmetic heads). The design is scalable and evaluation is thorough.

**Strengths:**

1. The paper clearly articulates the problem of "attention superposition" as a key obstacle to interpretability, drawing a compelling analogy to superposition in MLP layers. This provides a strong theoretical motivation for the work.

2. The design (rank-1 OV circuits, top-K sparsity, QK weight sharing) is effective and scalable.

3. Comprehensive experiments, including scaling laws, automated interpretability, and compelling case studies on models up to Llama-8B.

**Weaknesses:**

1. The primary limitation is that QK circuits are shared across heads, potentially hindering full disentanglement and causal attribution.

2. Evidence for novel head functionalities is mostly correlational; more causal intervention experiments would strengthen the claims.

**Questions:**

What might the unreconstructible "dark matter" in the attention output represent?

---

> ### Author Response · Authors · 2025-11-23
> **Potential solutions to QK sharing & More causal intervention experiments (1 / 2)**
>
> We are very grateful for your recognition and constructive suggestions on further improving this work.
>
> > The primary limitation is that QK circuits are shared across heads, potentially hindering full disentanglement and causal attribution.
>
> Thanks for pointing this out. Our current take on this is mostly discussed in Appendix A. To mitigate this, we are most optimistic about a hybrid approach of
> - Modifying our QK circuit to a mixture of shared parameters and an individual transformation (or adapter) for each head so that QK independence comes at a lower cost.
> - dynamically reducing dimension for Lorsa QK pairs, which is very analougous to [Mixture of Linear Transformation](https://transformer-circuits.pub/2025/bulk-update/index.html) using a variety of transformation ranks.
> - Some kind of post training to learn independent QK circuits with an acceptable number of training tokens.
>
> The current version of Lorsa is very preliminary and we expect to see significant modifications as this direction progresses. If you are interested in any of the modifications above or any suggestions you are excited about, we are willing to run the experiments in the upcoming discussion period.
>
>
> > Evidence for novel head functionalities is mostly correlational; more causal intervention experiments would strengthen the claims.
>
> Thanks for pointing this out. We only included a very brief case showing Lorsa $W_o$ can serve as steering vectors to causally affect outputs. Here we show 3 other cases.
>
> 1. For the prompt `calc:45+23=`, the model can successfully predict 68 (p=0.668). After we suppress `L15.13599: op2 in [20, 24]` and activate `L15.13535: op2 in [10, 14]` in layer 15, the model predicts 58 (p=0.652).
>
> 2.  For the prompt `calc:65-14=`, the model can successfully predict 51 (p=0.676). After we suppress `L15.13421: min(op1,op2) % 10 = 4` and activate `L15.13602: min(op1,op2) % 10 = 7` (i.e. Modifying "-14" to "-17") in layer 15, the model predicts 48 (p=0.211, top prob predicted).
>
> 3. A [golden-gate-claude-style](https://x.com/mlpowered/status/1792948227433775525) theme twist:
>   - Prompt: `Write a poem about death, focusing on mortality, loss, and the passage of time. Use vivid imagery and metaphors to evoke melancholy, contemplation, or awe. Explore death as an end, a transition, or a natural part of life, making the poem reflective and emotionally resonant.\nWrite only the poem in the following:`
>   - Clean output: Death is a journey, a destination, and a reminder of the brevity of life. It is a transition from one state to another, a passage from the physical to the spiritual. It is a time to reflect on our lives, to consider our choices and the legacy we leave behind. It is a reminder that life is precious and fleeting, and that we should make the most of every moment we have.
>   - Suppress: L15.14879, a theme anchor head attending to "death", "soul", "funeral", "frightening"
>   - Activate: L15.15201 attending to "dream", "magic", "sweet", "reality"
>   - Intervened output: Youth is the season of hope and dreams, but as we age, we gain wisdom and experience. Then life becomes a graceful, woven reference.
>
> These results are still far from a systematic investigation. Nonetheless, we believe similar steering effects of Lorsa and SAE features can be observed in a rigorous study due to their analogous linear nature. For now, we are unable to conduct such experiments due to time and resource limitations.

---

> ### Author Response · Authors · 2025-11-23
> **Discussions on "dark matters" (2 / 2)**
>
> > What might the unreconstructible "dark matter" in the attention output represent?
>
> The study of irreducible error in SAE scaling laws (i.e. dark matter) has been an open question in the field. [Engels et al.](https://arxiv.org/abs/2410.14670) finds that a proportion of irreducible error can be predicted linearly from the input, but failed to systematically explain its origin. Nonetheless, their explanatation that this might be dense linear features, and the residual was "Nonlinear error" makes intuitive sense.
>
> We have some very preliminary explorative experiments showing sparse dictionary learning methods, even trained on activations composed solely with linear, sparse features, fail to learn the true underlying features with small activation values or frequency. This might be due to the negligible supervision signal these features provided during training. [A related solution](https://aclanthology.org/2025.findings-naacl.87/) is training on a narrow distribution of data we are particularly interested in.
>
> Since language models themselves are not perfect, their activation might not be as well. For instance, attention heads may attend to totally unrelated tokens for no reason. And embeddings of rare tokens might be pure noise, etc. To make things worse, this noise will accumulate through layers and token positions. We conjecture this may be the main source of nonlinear error.
>
> We are currently unable to go further in this direction. But we think at the end of the day, it might not be necessary to fully explain activations to fully understand the model's behavior, as long as we are able to identify which part of dark matter does not require explaining.
>
> From a pragmatical perspective, there is still large space for improvement in Lorsa (as well as transcoders) architecture, initialization and training strategy to further push the scaling law of these replacement models so we can see more interesting behaviors of the model by simply scaling them up. But eventually the community will have to come back to the dark matter question.

---

### Official Review · Reviewer_um4H · 2025-11-01

**Soundness:** 3
**Presentation:** 3
**Contribution:** 3
**Rating:** 6
**Confidence:** 2

**Summary:**

The paper proposes a sparse replacement model of the attention layer,  Lorsa, to understand the attention   mechanisms. The authors demonstrate that Lorsa can rediscover previously reported mechanisms such as induction heads and attention sinks, and even identify new patterns such as subtoken induction heads.

**Strengths:**

- The proposed Lorsa method successfully rediscovers known attention mechanisms such as induction heads and attention sinks, and also discovers new interpretable phenomena (e.g. subtoken heads and arithmetic Lorsa-heads. I'm not very familiar with the literature, but the proposed approach appears to be novel.
- The paper is well organized and is easy to follow.

**Weaknesses:**

- The work may be better if including motivation and ablation study of several of the design choices, especially the difference with SAE, e.g. predicting the downstream activations, the top-K operation.

**Questions:**

- Is it possible to use existing methods (e.g. SAE) to identify the same newly discovered attention mechanisms reported in this paper?
- Could the authors provide more details on how well autointerp performs, and clarify whether it can be trusted as a quantitative metric for comparing interpretability across different methods in Section 5.3?
- How are gradients propagated through the Top-K operation during training?

---

> ### Author Response · Authors · 2025-11-23
> **Ablation Studies and Comparison with SAEs (1 / 2)**
>
> We are grateful for your constructive feedback and questions on several comparisons with SAEs
>
> > The work may be better if including motivation and ablation study of several of the design choices, especially the difference with SAE, e.g. predicting the downstream activations, the top-K operation.
>
> If we understood your suggestions properly, these would be adding comparison with SAEs used for predicting downstream tasks and justifying the use of topk activation function. There has been established liteature in the former topic, which was first introduced in [this work](https://arxiv.org/abs/2405.13868) and is currently known as [transcoders](https://arxiv.org/abs/2406.11944). The main motivation is that we can have a linear read-in component (encoder for transcoders and $W_v$ for Lorsa) to see how each feature is activated from upstream ones and how features contribute to downsteam tasks with decoders for transcoders and $W_o$ for Lorsa. So it becomes possible to connect the interpretable dots and see the 'true' computation graph given different prompts, albeit reconstruction errors.
>
> Ablating TopK sparsity is a reasonable concern. Possible alternatives include JumpReLU and BatchTopK. We suspect it will also be the case that BatchTopK will slightly outperform TopK, with JumpReLU falling behind in terms of reconstruction fidelity-sparsity pareto front. We did not include this ablation because of space limits and we think ablations on other design choices (e.g. QK dimension, scaling laws) can be more important for Lorsa. It might also be the case that our priori can be false for Lorsa. We are willing to include these ablations if you have further concerns about this.
>
> > Is it possible to use existing methods (e.g. SAE) to identify the same newly discovered attention mechanisms reported in this paper?
>
> Yes, we expect SAEs and Lorsas to produce similar features trained on the same attention layer. For instance, there is an [induction feature](https://www.neuronpedia.org/gemma-2-2b/14-gemmascope-att-16k/14804) specific to [Dr. xxx ... Dr.] which moves the name when "Dr." is present in layer 14 of gemma-2b. This is similar to our findings of Lorsa induction heads.

---

> ### Author Response · Authors · 2025-11-23
> **Autointerp & TopK Computation (2 / 2)**
>
> > Could the authors provide more details on how well autointerp performs, and clarify whether it can be trusted as a quantitative metric for comparing interpretability across different methods in Section 5.3?
>
> Our current take on autointerp is mixed: it can roughly tell whether a group of features is interpretable or not, but the precise numeric value might not be very informative. For the use case in this work, the result can be interpreted as 'these two methods produce roughly comparably interpretable features, with later layers less comprehensible' but Lorsa's 6 wins and 3 losses do not imply superior interpretability. We would like to give a brief literature review to support this.
>
> Positive evidence:
>
> 1. (Quantitive) In the [original autointerp paper](https://openaipublic.blob.core.windows.net/neuron-explainer/paper/index.html#sec-human-scoring), OpenAI researchers report an agreement rate of ~0.8 on human rating and autointerp scores. The experiment was conducted on raw neurons instead of SAE features, so the investigated top activations include both poly- and mono-semantic patterns.
>
> 2. (Might be subjective) Anthropic Interp Team leverages autointerp to present their results in their [SAE paper](https://transformer-circuits.pub/2023/monosemantic-features/vis/a1.html) and [circuit tracing paper](https://transformer-circuits.pub/2025/attribution-graphs/biology.html?slug=calc-36-plus-59#french-multilignual-big-svg). These results make intuitive sense for most features they studied, and autointerp scores often align with our intuition.
>
> 3. (Might be subjective) Neuronpedia Team provides autointerp for gemma scope and llama scope SAEs, which is often accurate for searching features we are interested in. ([Link](https://www.neuronpedia.org/gemma-2-2b/?sourceSet=gemmascope-mlp-16k&selectedLayers=[]&sortIndexes=[]&ignoreBos=true&q=Chef%20Ramirez%27s%20signature%20dish%2C%20a%20fusion%20of%20Thai%20and%20Mexican%20cuisine%2C%20features%20spicy%20green%20curry%20enchiladas%2C%20tantalizing%20the%20taste%20buds%20with%20a%20bold%2C%20innovative%20flavor%20combination.))
>
> Negative Evidence:
>
> 1. [This ICLR submission](https://openreview.net/forum?id=USyGD0eUod) states that autointerp metrics cannot tell between SAEs trained on randomly initialized transformers and pretrained models. [This work by Anthropic](https://transformer-circuits.pub/2023/monosemantic-features/index.html#global-analysis-about-model) reports similar results that SAEs for untrained transformers contain almost exclusively single-token features (which may due to the fact that for randomly initialized RoPE based transformers, residual stream activations are nearly a function of the current token plus some noise from attention). Our interpretation of these results is that these single token features contribute a lot to autointerp scores for untrained transformers since they are easier to understand and simulate. So it wont render autointerp completely useless.
>
> 2. It can be hard for LLMs to capture complex patterns like induction heads (for example, we pick induction lorsa heads with curated prompts rather than autointerp results). [This feature](https://www.neuronpedia.org/gemma-2-2b/14-gemmascope-att-16k/14804) in gemma scope attention SAE is a case where autointerp fails to recognize induction features.
>
> To briefly sum up, autointerp is not perfect. So we can only infer from our autointerp results that "at least SAEs and Lorsas provide interpretable results in general". However, this might be the best metric at hand to have an overall idea of interpretability of thousands or millions of features for now.
>
> > How are gradients propagated through the Top-K operation during training?
>
> Only selected channels (features) receive gradient from Top-K, i.e.
> ```
>     class TopK(torch.autograd.Function):
>         @staticmethod
>         def forward(ctx, x, k=2):  # e.g. x = [1, 4, 5, 2, 3]
>             ctx.save_for_backward(get_binary_mask_of_top_k_indices(x, k))  # [0, 1, 1, 0, 0]
>             return top_k(x, k)  # [0, 4, 5, 0, 0]
>
>         @staticmethod
>         def backward(ctx, grad_output):
>             top_k_idx_binary_mask = ctx.saved_tensors
>             return grad_output * top_k_idx_binary_mask
> ```
>
> We thank you again for your valuable feedback and look forward to more discussions.

---

### Author Response · Authors · 2025-12-03
**Thanks for the reviewers' suggestions & a brief summary of the discussion**

We understand that the discussion period ended due to sudden reasons, which lead to insufficient further discussions of this submission. We would like to briefly highlight some important feedbacks.

Reviewer g9Wx and Reviewer ju5a both questioned about what the unreconstructible part of attention output represent. Our thoughts on this open problem are limited to some hypotheses and possible explanations based on existing literature. Our work shows that reconstruction errors of attention replacement models and attention SAEs are very similar in terms of norm and direction. But we are unable to go further in this direction for now.

Reviewer 8qJU expressed concerns on the mechanistic faithfulness of "replacement models", which is also an important open question. Mere reconstruction does not necessarily lead to mechanistic faithfulness of the underlying model. The replacement model paradigm largely relies on the hypothesis that the "algorithms" implemented by attention can be (largely) described via an over-complete set of sparsely activating low-dim linear attentional units (i.e. attention superposition). We provide more clarification in the revised manuscript and a proof to bound mechanistic faithfulness based on the feedbacks of Reviewer 8qJU.

We are also grateful for other suggestions by all reviewers in terms of Lorsa head causality (Reviewer ju5a), relation to SAEs (Reviewer um4H), reliability of autointerp (Reviewer um4H), important notation issues & typos (Reviewer g9Wx) and some important phrasing risks for non-interp readership (Reviewer 8qJU). We tried to address each concern in the corresponding comments and modified the manuscript where applicable.

There is much space for improvement in Lorsa architecture. This work aims to show that attention superposition can be resolved to a large extent with attention replacement models like Lorsa.  And we chose to focus on ablations on crucial architectural designs, especially for Lorsa QK circuits (Appendix B) to facilitate future work.

---

### Meta-Review · Area_Chair_thnC · 2026-01-06

**Summary:**

## Summary
This paper proposes Low-Rank Sparse Attention (Lorsa), a sparse replacement model of Transformer attention layers that approximates and decomposes multi-head self-attention (MHSA) into individually interpretable low-rank components. Lorsa can find previously known heads such as induction heads, successor heads, attention sinks, and those handling arithmetic tasks, and it identifies novel types called subtoken induction heads and theme anchors.

## Reviewer Concerns
Major concerns raised by reviewers can be summarized as follows.
- **Ablation study**. Reviewer un4H asked for more ablation studies of design choices, especially focusing on the difference from SAE.
- **Reliability of autointerp**. Reviewer un4H questioned how well autointerp performs and whether it can be trusted.
- **Shared QK**. Reviewer ju5a mentioned shared QK as one of the limitations, which was also noted by the authors in the paper.
- **Correlation vs causation**. Reviewer ju5a pointed out that novel head functionalities are mostly correlational, and more causal intervention experiments are required.
- **Dark matter**. Reviewers ju5a and g9Wx asked what constitutes the unreconstructible "dark matter”, and what would be the implications for the completeness of the discovered units.
- **Confusing notation**. Reviewer g9Wx noted that some notation in the paper is confusing and inconsistent.
- **Positioning**. Reviewer 8qJU expressed concern about the paper positioning Lorsa as “a replacement model,” and suggested that the method should be framed as an interpretability tool.
- **Approximation vs behavior**. Reviewer 8qJU commented that approximating MHSA output may not directly correspond to models having the same behavior.

**Reviewer Concerns:**

Unfortunately, the discussion period closed before we received feedback from three out of the four reviewers. Based on the rebuttal, the authors’ responses can be summarized as follows.

- **Ablation study**. The authors clarified the connection with SAE and the reason why ablations on TopK sparsity was not included due to space limits. This ablation would strengthen a future revision, in my judgment.

- **Reliability of autointerp**. The authors discussed the evidence for and against the trustworthiness of autointerp, with concrete examples of prior results.

- **Shared QK**. The authors discussed some possible remedies, while admitting that the current version of Lorsa is very preliminary and further follow-up study is needed.

- **Correlation vs causation**. Some preliminary causal intervention experiments were provided, suggesting the plausibility of the newly discovered head functionalities.

- **Dark matter**. The authors discussed the investigation of the “dark matter” in the literature, and shared their thoughts on its origin.

- **Confusing notation**. The authors clarified their notation and updated the manuscript. However, $D_{QK}^{Lorsa}$ is still first used without definition, as Algorithm 1 uses another symbol $d_h$. The symbol used in the authors’ rebuttal also does not fully match the paper. On a related note, I suspect that $d_{head}^{QK}$ in the discussion section is an outdated version that requires an update.

- **Positioning**. The authors clarified that Lorsa is primarily an interpretability tool, and corrected confusion around the term “a replacement model.”

- **Approximation vs behavior**. The authors added a simple analysis (albeit with strong assumptions) that the reviewer suggested. They also added an honest comment that despite the bound it is still possible that replacement models are not strictly mechanistically faithful to the underlying module, and that this issue is shared by many replacement model approaches.

**Reviewer Scores:**

The initial reviews had scores 8/6/6/4, which was already quite positive. Only Reviewer g9Wx (initial score 8) was able to respond to the rebuttal before the discussion phase closed; they maintained their score.

I can see that the authors put a lot of effort into the rebuttal, trying to respond to every bit of reviewers’ comments and questions, in some cases offering a good summary of the current status of the literature. Some of the criticisms (reliability of autointerp, faithfulness of replacement model, etc) on the paper were ones shared by the emerging literature, and hence understandable. I expect that one or two reviewers may have raised their score.

I believe the proposed method, though it is preliminary, has a good potential to be developed to make a useful tool for identifying various functionalities of attention modules. With overall good review scores and well-addressed concerns, I incline toward acceptance.

In my opinion, the paper would benefit from a thorough revision to make it more accessible to readers outside the mechanistic interpretability community. SAEs serve as a fundamental baseline in this paper, but for readers that do not have a good prior knowledge on the use of SAEs in the community, it can be difficult to capture the core distinction and advantage of Lorsa relative to SAEs.

---

### Decision · Program_Chairs · 2026-01-26

Accept (Poster)